# Solving a Special Type of Optimal Transport Problem by a Modified Hungarian Algorithm

**Yiling Xie**                                                                  *yxie350@gatech.edu*
*School of Industrial and Systems Engineering*
*Georgia Institute of Technology*

**Yiling Luo**                                                                  *yluo373@gatech.edu*
*School of Industrial and Systems Engineering*
*Georgia Institute of Technology*

**Xiaoming Huo**                                                                *huo@gatech.edu*
*School of Industrial and Systems Engineering*
*Georgia Institute of Technology*

**Reviewed on OpenReview:** *https://openreview.net/forum?id=k5m8xXTOrC*

## Abstract

Computing the empirical Wasserstein distance in the Wasserstein-distance-based independence test is an optimal transport (OT) problem with a special structure. This observation inspires us to study a special type of OT problem and propose *a modified Hungarian algorithm* to solve it *exactly.* For the OT problem involving two marginals with $m$ and $n$ atoms ($m \geq n$), respectively, the computational complexity of the proposed algorithm is $\mathcal{O}(m^2 n)$. Computing the empirical Wasserstein distance in the independence test requires solving this special type of OT problem, where $m = n^2$. The associated computational complexity of the proposed algorithm is $\mathcal{O}(n^5)$, while the order of applying the classic Hungarian algorithm is $\mathcal{O}(n^6)$. In addition to the aforementioned special type of OT problem, it is shown that the modified Hungarian algorithm could be adopted to solve a wider range of OT problems. Broader applications of the proposed algorithm are discussed—solving the one-to-many assignment problem and the many-to-many assignment problem. We conduct numerical experiments to validate our theoretical results. The experiment results demonstrate that the proposed modified Hungarian algorithm compares favorably with the Hungarian algorithm, the well-known Sinkhorn algorithm, and the network simplex algorithm.

## 1  Introduction

One appealing application of optimal transport (OT) and Wasserstein distance (Villani, 2009; Peyré & Cuturi, 2019) is the independence test. The Wasserstein distance between two distributions $\mu_1, \mu_2$ on $Z$ is defined as:

$$W(\mu_1, \mu_2) := \inf \left\{ \int_{Z^2} d(z, z') d\gamma(z, z') : \ \gamma \text{ is a distribution with marginals } \mu_1 \text{ and } \mu_2 \right\},$$

where $(Z, d)$ is a metric space (1-Wasserstein distance is considered in this paper). The Wasserstein distance is a metric on probability measures (Villani, 2009). To test the independence between the variables $Y \sim \nu_1$ and $Z \sim \nu_2$, people utilize the Wasserstein distance between the joint distribution of $Y, Z$ and the product distribution of $Y, Z$, i.e., $W(\pi, \nu_1 \otimes \nu_2)$, where $\pi$ denotes the joint distribution of $Y, Z$, and $\nu_1 \otimes \nu_2$ denotes the product distribution of $Y, Z$. While the statistical properties of this approach have been intensely investigated (Nies et al., 2021; Mordant & Segers, 2022; Wiesel, 2022), no existing literature focuses on the computational aspect. In this paper, we discuss the following:

*How to compute the empirical Wasserstein distance in the independence test?*

In practice, given $n$ i.i.d. samples $\{(y_1, z_1), \cdots, (y_n, z_n)\}$ generated from $(Y, Z)$, one can build the statistic—$W(\widehat{\pi}, \widehat{\nu}_1 \otimes \widehat{\nu}_2)$, where $\widehat{\pi}, \widehat{\nu}$ denote the corresponding empirical distributions of $\pi$ and $\nu$, respectively—to test the independence. Computing $W(\widehat{\pi}, \widehat{\nu}_1 \otimes \widehat{\nu}_2)$ is equivalent to solving the following optimization problem: (more details are presented in Section 4.)

$$\min_{X^\circ \in \Pi^\circ} \sum_{i,j,k=1}^{n} d((y_i, z_j), (y_k, z_k)) X^\circ_{ij;k}, \ \Pi^\circ = \left\{ X^\circ_{ij;k} \geq 0 \middle| \sum_{k=1}^{n} X^\circ_{ij;k} = \frac{1}{n^2}, \sum_{i,j=1}^{n} X^\circ_{ij;k} = \frac{1}{n}, \forall i, j, k = 1, \cdots, n. \right\},$$

$$(1)$$

where the metric $d$ is usually chosen as $d((y_i, z_j), (y_k, z_l)) = \|y_i - y_k\|_p + \|z_j - z_l\|_p$, and $\| \cdot \|_p$ denotes the $l_p$ norm.

Problem (1) is an OT problem involving two marginals. One marginal is uniform with $n$ atoms (i.e., we have $\sum_{i,j=1}^{n} X^\circ_{ij;k} = 1/n, \forall k, 1 \leq k \leq n$), and the other marginal is uniform with $n^2$ atoms (i.e., we have $\sum_{k=1}^{n} X^\circ_{ij;k} = 1/n^2, \forall i, j, 1 \leq i, j \leq n$). Motivated by this structure, we study the following special OT problem:

$$\min_{X' \in \mathcal{U}'} \sum_{i=1}^{m} \sum_{j=1}^{n} X'_{ij} C_{ij}, \quad \mathcal{U}' = \left\{ X'_{ij} \geq 0 \middle| \sum_{j=1}^{n} X'_{ij} = \frac{1}{m}, \sum_{i=1}^{m} X'_{ij} = \frac{m_j}{m}, \forall i = 1, \cdots, m; j = 1, \cdots, n \right\}. \quad (2)$$

where $0 < n \leq m$, $m_j$'s are positive integers, and $\sum_{j=1}^{n} m_j = m$ holds. One marginal of this OT problem is $n$-dimensional where the probability of each component is prescribed as $m_j/m$ (i.e., we have $\sum_{i=1}^{m} X'_{ij} = m_j/m, \forall j, 1 \leq j \leq n$), and the other marginal is uniform with $m$ atoms (i.e., we have $\sum_{j=1}^{n} X'_{ij} = 1/m, \forall i, 1 \leq i \leq m$). In essence, problem (1) is a special case of problem (2), where $m_j = n, m = n^2, \forall j = 1, \cdots, n$. Throughout this paper, we consider real-valued entries in the cost matrix and later propose a strongly polynomial-time algorithm to solve problem (2) precisely.

Per Birkhoff's theorem (Birkhoff, 1946), the solution to problem (2) is a vertex (whose coordinates are zeros and ones). Then, we could prove the following proposition, and the proof is relegated to the Appendix.

**Proposition 1.** *The optimization problem (2) is equivalent to the optimization problem (3).*

$$\min_{X \in \mathcal{U}} \sum_{i=1}^{m} \sum_{j=1}^{n} \frac{1}{m} X_{ij} C_{ij}, \quad \mathcal{U} = \left\{ X_{ij} = \{0, 1\} \middle| \sum_{j=1}^{n} X_{ij} = 1, \sum_{i=1}^{m} X_{ij} = m_j, \forall i = 1, \cdots, m; j = 1, \cdots, n \right\}. \quad (3)$$

One may recall the assignment problem, seeing the definition in Section 2, where the permutation matrix is the solution matrix. $X \in \mathcal{U}$ is similar but different from the permutation matrix: $X \in \mathcal{U}$ is an $m \times n$ matrix instead of a square matrix and has *multiple* entries of 1 in each column instead of only one entry. In this case, we are not able to directly apply algorithms for the assignment problem, such as the Hungarian algorithm (Kuhn, 1955; Munkres, 1957). An approach to obtain the precise solution to problem (3) is first to duplicate the columns of $C$ and $X$, then apply the Hungarian algorithm. The computational complexity of this approach is $\mathcal{O}(m^3)$. In this paper, *a modified Hungarian algorithm* is proposed. The algorithm specializes in solving the special type of OT problem (3), which is equivalent to problem (2), with a provable lower order—$\mathcal{O}(m^2 n)$.

For the special type of OT problem (2), we require that one marginal of the OT problem should be uniform. We could further relax the uniform requirement and consider the following more general OT problems:

$$\min_{X^* \in \mathcal{U}^*} \sum_{i=1}^{m} \sum_{j=1}^{n} X^*_{ij} C_{ij}, \quad \mathcal{U}^* = \left\{ X^*_{ij} \geq 0 \middle| \sum_{j=1}^{n} X^*_{ij} = \frac{n_i}{M}, \sum_{i=1}^{m} X^*_{ij} = \frac{m_j}{M}, \forall i = 1, \cdots, m; j = 1, \cdots, n \right\}, \quad (4)$$

where $0 < n \leq m$, $n_i$'s, $m_j$'s are positive integers, and $\sum_{j=1}^{n} m_j = \sum_{i=1}^{m} n_i = M$ holds. The modified Hungarian algorithm could be adapted to solve problem (4), and the associated computational complexity is $\mathcal{O}(M^2 n)$.

Back to the Wasserstein-distance-based independence test problem (1), the resulting computational complexity of applying the proposed algorithm is $\mathcal{O}(n^5)$ while the order of applying the classic Hungarian algorithm is $\mathcal{O}(n^6)$. In this sense, the proposed algorithm is faster. In addition to the application in the Wasserstein independence test, broader applications of the modified Hungarian algorithm, including solving the one-to-many assignment problem and the many-to-many assignment problem (Zhu et al., 2011; 2016), are investigated. Two practical assignment problems involving the soccer game and agent-task assignment serve as examples to illustrate how to apply the proposed algorithm.

## 1.1 Related work

### 1.1.1 Semi-assignment problem

The problem (3) is also called the semi-assignment problem in the literature (Barr et al., 1977; Kennington & Wang, 1992). Kennington & Wang (1992) proposes a strongly polynomial-time to solve the semi-assignment problem exactly. The proposed modified Hungarian algorithm and the algorithm proposed in Kennington & Wang (1992) are fundamentally different. The algorithm in Kennington & Wang (1992) adjusts the shortest path augmenting algorithm (Jonker & Volgenant, 1987), while our algorithm modifies the Hungarian algorithm Kuhn (1955). As illustrated in Jonker & Volgenant (1987); Kennington & Wang (1992), the Hungarian algorithm is a primal-dual method based on maximum flow, while the shortest path augmenting algorithm considers the assignment problem as a minimum cost flow problem and is a dual method based on the shortest path. More specifically, the modified Hungarian algorithm is based on a modified Kuhn-Munkres theorem and iterates to identify a perfect pseudo-matching in the bipartite graph with some feasible dual variables, while the algorithm in Kennington & Wang (1992) involves constructing the shortest augmenting path in the auxiliary graph, and the flow is pushed along the path (Kennington & Wang, 1992). Regarding computational complexity, both algorithms have the order of $\mathcal{O}(m^2 n)$. However, our algorithm is easier to understand and implement because there are four phases (column reduction, reduction transfer, row reduction augmentation, and shortest path augmentation) in the algorithm in Kennington & Wang (1992) while our algorithm only involves two phases (updating the feasible labeling and improving the pseudo-matching).

### 1.1.2 Approximation algorithms

To solve the OT problem, we could apply the exact algorithms or the approximation algorithms. While the proposed modified Hungarian algorithm is an exact OT solver, there are a bunch of approximation algorithms. People usually consider the following OT problem:

$$\min_{X \in \mathcal{U}(\alpha,\beta)} \sum_{i=1}^{N_1} \sum_{j=1}^{N_2} X_{ij} C_{ij}, \quad \mathcal{U}(\alpha,\beta) := \left\{ X \in \mathbb{R}_+^{N_1 \times N_2} \left| \sum_{i=1}^{N_1} X_{ij} = \alpha_j, \sum_{j=1}^{N_2} X_{ij} = \beta_i \right. \right\}, \tag{5}$$

where $\sum_{i=1}^{N_1} \beta_i = 1$ and $\sum_{j=1}^{N_2} \alpha_j = 1$. The approximation algorithms (Cuturi, 2013; Dvurechensky et al., 2018; Lin et al., 2019a; Xie et al., 2022) in the literature are to obtain an $\epsilon$-approximation $\widehat{X} \in \mathcal{U}(\alpha,\beta)$ to (5) such that $\langle \widehat{X}, C \rangle \leq \langle X^*, C \rangle + \epsilon$, where $X^*$ is the solution to (5).

The development of efficient exact algorithms is meaningful. Notably, precise solutions are needed in some scenarios, and Dong et al. (2020) demonstrates the favorable numerical performance of the exact solutions over the approximate solutions. Numerical experiments are conducted to compare the modified Hungarian algorithm with the most widely-used approximation algorithm—the Sinkhorn algorithm, highlighting the efficiency of our exact algorithm.

### 1.1.3 Exact algorithms

We review the exact algorithms to solve the OT problem and compare our algorithm with them as follows.

The special type of OT problem (2) is a minimum-cost flow problem and could be solved by the network simplex algorithms. Note that Orlin (1997) proposes the first polynomial-time network simplex algorithm, and Tarjan (1997) further improves the result. The associated computational complexity of applying Tarjan's

algorithm to problem (2) is $\mathcal{O}(m^2 n \log(m) \min\{\log(mC_{max}), mn \log(m)\})$, where $C_{max}$ denotes the maximum absolute value of the costs if all costs are integers and $\infty$ otherwise (Orlin, 1997; Tarjan, 1997). More specifically, if the costs are integral, the resulting computational complexity is $\mathcal{O}(m^2 n \log(m) \log(mC_{max}))$, which is comparable to the proposed algorithm; if the costs are not integral, the resulting computational complexity is $\mathcal{O}(m^3 n^2 \log^2(m))$, which is worse than our algorithm.

The interior point algorithms can also be customized to solve the minimum-cost flow problems, e.g., Yeh (1989); Resende & Veiga (1993). As discussed in Resende & Pardalos (1996), from the perspective of computational complexity, Vavasis & Ye (1994) proposes a strongly polynomial-time interior point algorithm for solving the minimum-cost flow problem. The adaption of this method to problem (2) has the order of $\mathcal{O}(m^{6.5} n^{6.5} \log(mn))$, which is much worse than the proposed algorithm. As demonstrated in Resende & Pardalos (1996), prior to Vavasis & Ye (1994), the fastest interior point method to solve the minimum-cost flow problem comes from Vaidya (1989). The computational complexity of the adoption of Vaidya's algorithm to problem (2) is $\mathcal{O}(m^{2.5} n^{0.5} \log(mC_{\max}))$. The complexity is worse than our algorithm, and the algorithm requires that all costs are integers.

As shown in Peyré & Cuturi (2019), we could apply other combinatorial algorithms, including the auction algorithm (Bertsekas & Eckstein, 1988), and the dual ascent algorithm (Bertsimas & Tsitsiklis, 1997), to solve the special type of OT problem (2). We compare the proposed modified Hungarian algorithm with them as follows: Regarding the auction algorithm, the output is suboptimal (Peyré & Cuturi, 2019), while the output of our algorithm is optimal. Furthermore, Bertsimas & Tsitsiklis (1997); Bertsekas (1988) illustrate that finite termination of the auction algorithm with an optimal solution could come from the nature of integer-valued cost coefficients. In terms of the dual ascent algorithm, Bertsimas & Tsitsiklis (1997) demonstrates that the input of integral costs is one of the conditions for finite termination.

In conclusion, finite termination with an optimal solution or fast computation of the aforementioned algorithms, including network simplex algorithm, interior point algorithm, auction algorithm, and dual ascent algorithm, require integer-valued costs, while the proposed modified Hungarian algorithm could handle real-valued costs. Although most of the problems in practice have rational costs, which can be scaled to integer-valued costs, it is more convenient to use the proposed algorithm since it allows direct input of any real-valued costs. Also, looking into OT problems with real-valued costs itself is of much theoretical interest.

Similar discussions and comparisons could be developed for the more general OT problem (4). Problem (4) could be reformulated to a minimum-cost flow problem and can be solved by the exact solvers mentioned above. Among the algorithms, the computational complexity of the network simplex algorithm can be comparable to the proposed modified Hungarian algorithm—$\tilde{\mathcal{O}}(M^2 n)$—under the assumption that all costs are integral.

### 1.1.4 Independence criteria

There are some other independence criteria based on OT or the Wasserstein distance. Shi et al. (2020); Deb & Sen (2021) design the independence criterion by combining the distance covariance and OT. Liu et al. (2022) applies the entropy-regularized OT. Wiesel (2022) utilizes the nested Wasserstein distance. The criterion proposed in Mordant & Segers (2022) is based on the 2-Wasserstein distance under the quasi-Gaussian assumption. This paper considers the criterion based on the general Wasserstein distance as shown in the formulation (1).

### 1.2 Our contributions:

We propose a modified Hungarian algorithm to solve a special type of OT problem (2). The modification enables us to deal with the scenario where two marginals have different sizes of atoms, and the atoms in one of the marginals have multiple assignments. Further, the proposed modified Hungarian algorithm could be extended to solve more general OT problems. Moreover, the applications of the proposed algorithm are explored: adopting the modified Hungarian algorithm to solve the Wasserstein independence test problem (1), the one-to-many assignment problem, and the many-to-many assignment problem. Finally, several

numerical experiments are carried out using Python to show the favorability of our algorithm over the classic Hungarian algorithm, the Sinkhorn algorithm, and the network simplex algorithm.

### 1.3 Organization

The remainder of this paper is organized as follows. In Section 2, we introduce some basics of graph theory. In Section 3, we propose the modified Hungarian algorithm and compute its computational complexity. In Section 4, we apply the modified Hungarian algorithm to the Wasserstein-distance-based independence test problem. In Section 5, we apply the modified Hungarian algorithm to the one-to-many assignment problem and the many-to-many assignment problem. In Section 6, we carry out various numerical experiments on both synthetic data and real data to validate our theoretical results and show the favorability of our algorithm. We discuss some future work in Section 7.

## 2 Preliminaries

Some definitions related to combinatorial optimization and graph theory (Ahuja et al., 1988; Suri, 2006; Burkard et al., 2012) are introduced. They will be needed in the rest of this paper.

**Definition 1** (Assignment problem). *Given an $k \times k$ cost matrix with components $c_{ij} \geq 0, i, j \in [k] = \{1, \cdots, k\}$, the assignment problem is to solve $\min_\phi \sum_{i=1}^k c_{i\phi(i)}$, where $\phi$ is the permutation of set $[k]$.*

**Definition 2** (Bipartite graph). *A graph $G = (V, E)$ is called a bipartite graph if its nodes can be partitioned into two subsets $V_1$ and $V_2$ so that for each edge $(v_1, v_2)$ in $E$, $v_1 \in V_1$ and $v_2 \in V_2$.*

**Definition 3** (Matching and perfect matching). *A matching in the bipartite graph $G = (V, E)$ is a subset $M \subset E$ such that at most one edge in $M$ is incident upon $v$, $\forall v \in V$. $M$ is called a perfect matching if every node in $G$ coincides with exactly an edge of $M$.*

**Definition 4** (Weighted bipartite graph). *A weighted bipartite graph is a bipartite graph where each edge has a weight $w(\cdot) \geq 0$. The weight of a matching $M$ is the sum of the weights of edges in $M$, i.e., $\sum_{e \in M} w(e)$.*

**Definition 5** (Labeling and feasible labeling). *For a weighted bipartite graph $G = (V, E)$, where $V = V_1 \cup V_2$, a labeling is a function $l : V \to \mathbb{R}$. A feasible labeling is one labeling such that $l(v_1) + l(v_2) \geq w(v_1, v_2), \forall v_1 \in V_1, v_2 \in V_2$.*

**Definition 6** (Equality graph and neighbor). *The equality graph w.r.t. the labeling $l$ is $G' = (V, E_l)$ where $E_l = \{(v_1, v_2) : l(v_1) + l(v_2) = w(v_1, v_2)\}$. The neighbor of $v_2 \in V_2$ and $S \subset V_2$ is defined as $N_l(v_2) = \{v_1 : (v_1, v_2) \in E_l\}$ and $N_l(S) = \cup_{v_2 \in S} N_l(v_2)$, respectively.*

**Definition 7** (Alternating and augmenting path). *Let $M$ be a matching of the bipartite graph $G = (V, E)$. A path in $G = (V, E)$ is a sequence of distinct nodes and edges $i_1, (i_2, i_2), \cdots, (i_{r-1}, i_r), i_r$, satisfying $(i_k, i_{k+1}) \in E$ for each $k = 1, \cdots, r-1$. A path in $G = (V, E)$ is alternating if its edges alternate between $M$ and $E - M$. An alternating path is augmenting if both endpoints do not coincide with any edges in $M$.*

## 3 Modified Hungarian algorithm

In this section, we propose a modified Hungarian algorithm to solve the special type of OT problem (2), which is equivalent to problem (3).

### 3.1 Review of the Hungarian algorithm

We first review the Hungarian algorithm (Kuhn, 1955; Munkres, 1957). Recall that the assignment problem is to solve $\min_\phi \sum_{i=1}^m c_{i\phi(i)}$. If we negate the costs and add the maximum of the costs to each component, solving the assignment problem is equivalent to finding a maximum weighted matching in the weighted bipartite graph with weights $w(i, j) = \max_{ij} c_{ij} - c_{ij}$. The Kuhn-Munkres theorem (Munkres, 1957) shows that finding a maximum weighted matching is equivalent to finding a perfect matching on the equality graph associated with some feasible labeling in the bipartite graph. In this regard, the Hungarian algorithm solves the assignment problem by identifying a perfect matching on some equality graph in the weighted bipartite

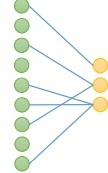 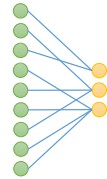

Figure 1: pseudo-matching (left) and perfect pseudo-matching (right), where $n = 3, m = 9, m_1 = 2, m_2 = 3, m_3 = 4$.

graph. One first generates an initial feasible labeling and an associated equality graph; then proceeds to look for an augmenting path to augment the matching. (if an augmenting path does not exist, update the feasible labeling.) If the matching is perfect on the equality graph concerning some feasible labeling, stop and output the matching.

### 3.2   Pseudo-matching

In problem (3), $X \in \mathcal{U}$ has one entry of 1 in each row, multiple entries of 1 in each column, and $0's$ elsewhere. Since a permutation matrix corresponds to a (perfect) matching in the bipartite graph, we define 'pseudo-matching' in the bipartite graph $G = (V_1 \cup V_2, E)$ to describe $X$. $V_1$ has $m$ nodes representing the rows of $X$ while $V_2$ has $n$ nodes representing the columns of $X$. Notice that we usually have $m > n$. In this case, each node in $V_1$ coincides with at most one edge, while multiple edges are allowed to connect with nodes in $V_2$. See the formal definition in Definition 8.

**Definition 8** (pseudo-matching, perfect pseudo-matching). *In the bipartite graph $G$, where $|V_1| = m, |V_2| = n$. $PM \subset E$ is a pseudo-matching if every node of $V_1$ coincides with at most one edge of $PM$, and jth node of $V_2$ coincides with at most $m_j$ edges of $PM$, where $\sum_{j=1}^{n} m_j = m$. Furthermore, if every node of $V_1$ coincides with exactly one edge of $PM$ and jth node of $V_2$ coincides with exactly $m_j$ edges of $PM$, $PM$ is called a perfect pseudo-matching.*

Figure 1 is an example of (perfect) pseudo-matching, where $n = 3, m = 9, m_1 = 2, m_2 = 3, m_3 = 4$. Under this setting, each node in the left-hand side of the graph can coincide with at most one edge, while each node in right-hand side can coincide with at most 2 edges, 3 edges, and 4 edges, respectively.

### 3.3   Our algorithm

Solving problem (3) is equivalent to looking for a maximum weighted pseudo-matching in the bipartite graph. We develop a modified Kuhn-Munkres theorem based on the pseudo-matching. See Theorem 1. (The proof can be found in the Appendix.) It demonstrates that we only need to find a perfect pseudo-matching on some equality graph to solve problem (3).

**Theorem 1** (Modified Kuhn-Munkres theorem). *If $l$ is a feasible labeling on the weighted bipartite graph $G = (V, E)$, and $PM \subset E_l$ is a perfect pseudo-matching on the corresponding equality graph $G' = (V, E_l)$, $PM$ is a maximum weighted pseudo-matching.*

Equipped with the modified Kuhn-Munkres theorem, we design a modified Hungarian algorithm (Algorithm 1). The definitions used in the algorithm are specified in Definition 9. The modified Hungarian algorithm improves either the feasible labeling (adding edges to the associated equality graph) or the pseudo-matching until the pseudo-matching is perfect on some equality graph w.r.t. some feasible labeling. The algorithm improves the pseudo-matching by generating pseudo-augmenting paths and then exchanging the edge status along the paths. This process is called the pseudo-augmenting process. Also, we force the pseudo-augmenting paths emanating from $V_2$, which have a lower order of nodes.

**Definition 9** (Free, matched, pseudo-matched, pseudo-alternating path, pseudo-augmenting path). *Let $PM$ be a pseudo-matching of $G = (V, E)$.*

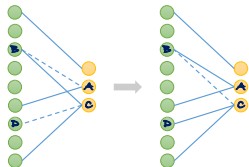

Figure 2: pseudo-augmenting process, where $n = 3, m = 9, m_1 = 2, m_2 = 3, m_3 = 4$.

- *If the node $v$ is in $V_1$, it is pseudo-matched if it is an endpoint of some edge in $PM$; if the node $v$ is the $j$th node in $V_2$, it is pseudo-matched if it is an endpoint of $m_j$ edges in $PM$. Otherwise, the node is free.*
- *If the node $v \in V$, we say it is matched if it is an endpoint of some edge in $PM$.*
- *A path is pseudo-alternating if its edge alternates between $PM$ and $E - PM$. A pseudo-alternating path is pseudo-augmenting if both its endpoints are free.*

An example of the pseudo-augmenting process is given in Figure 2. The solid line means that the edge belongs to the pseudo-matching. The dashed line means that the edge belongs to the equality graph but does not belong to the pseudo-matching. Node B and node C are pseudo-matched. Edge A-B and edge C-D are not in the pseudo-matching. In this sense, A-B-C-D is a pseudo-alternating path. Because node A and node D are free, A-B-C-D is a pseudo-augmenting. The pseudo-augmenting process is to exchange the status of the edges: delete B-C from the pseudo-matching and enter A-B, C-D into the pseudo-matching. The pseudo-matching has been improved in this way.

### 3.4 Computational complexity

We now analyze the computational complexity of Algorithm 1. Similar to the Hungarian algorithm (Suri, 2006), we keep track of $slack_{v_1} = \min_{v_2 \in S}\{l(v_1) + l(v_2) - w(v_1, v_2)\}$, $\forall v_1 \notin T$. The computational cost increases when computing $\alpha_l$ via $slack$s, updating the values of $slack$s, and calculating the labeling.

The number of edges of the pseudo-matching increases by 1 after one loop, so $\mathcal{O}(m)$ loops is needed to form a perfect pseudo-matching. There are two subroutines in each loop: the first is to update the feasible labeling (Step 2), and the second is to improve the pseudo-matching (Step 3). In the procedure of updating the feasible labeling, since there are $n$ nodes in $V_2$, the improvement occurs $\mathcal{O}(n)$ times to build a pseudo-alternating tree. In each time, computing $\alpha_l$, updating the $slack$s, and calculating the labeling cost $\mathcal{O}(m)$. In the procedure of improving the pseudo-matching, when a new node has been added to $S$, it costs $\mathcal{O}(m)$ to update $slack$s, and $\mathcal{O}(n)$ nodes could be added. On the other hand, when a node has been added to $T$, we just remove the corresponding $slack_{v_1}$. We conclude that each loop costs $\mathcal{O}(mn)$, so the total computational complexity of Algorithm 1 to solve problem (3) is $\mathcal{O}(m^2n)$. We summarize the analysis above in Theorem 2.

**Theorem 2.** *The computational complexity of applying the modified Hungarian algorithm to solve problem (3) is $\mathcal{O}(m^2n)$.*

### 3.5 Solving more general OT problem

We could adapt the proposed modified Hungarian algorithm to solve the more general OT problem (4). We first rewrite problem (4) as the formulation of the special type of OT problem (2) by duplicating the rows of the cost matrix, seeing Proposition 2 (the proof is relegated to the Appendix).

**Proposition 2.** *Problem (4) is equivalent to the following optimization problem:*

$$\min_{X^\dagger \in \mathcal{U}^\dagger} \sum_{i=1}^{M}\sum_{j=1}^{n} X_{ij}^\dagger C_{ij}^\dagger, \quad \mathcal{U}^\dagger = \left\{ X_{ij}^\dagger \geq 0 \middle| \sum_{j=1}^{n} X_{ij}^\dagger = \frac{1}{M}, \sum_{i=1}^{M} X_{ij}^\dagger = \frac{m_j}{M}, \forall i = 1, \cdots, m; j = 1, \cdots, n \right\}. \quad (6)$$

---

**Algorithm 1:** Modified Hungarian Algorithm

---

Generate an initial feasible labeling $l$: $\forall v_2 \in V_2, l(v_2) = 0; \forall v_1 \in V_1, l(v_1) = \max_{v_2 \in V_2}\{w(v_1, v_2)\}$ and initialize a pseudo-matching $M$ in $E_l$;

**1 if** *M is a perfect pseudo-matching* **then**
   | Stop
   **else**
   | Pick up a free node $v_{\text{free}} \in V_2$. Set $S = \{v_{\text{free}}\}, T = \emptyset$;
   | **for** $\mathbf{v_1} \in V_1$ *is matched to* $v_{free}$ **do**
   | | $T = T \cup \mathbf{v_1}$
   | **end**
   **end**
**2 if** $N_l(S) - T = \emptyset$ **then**
   | update labeling such that forcing $N_l(S) - T \neq \emptyset$:

   $$\alpha_l = \min_{v_1 \notin T, v_2 \in S}\{l(v_1) + l(v_2) - w(v_1, v_2)\}, \quad l(v) = \begin{cases} l(v) - \alpha_l & v \in S \\ l(v) + \alpha_l, & v \in T \\ l(v), & \text{otherwise} \end{cases}.$$

   **end**
**3 if** $N_l(S) - T \neq \emptyset$ **then**
   | pick $v_1 \in N_l(S) - T$;
   | **if** $v_1$ *is free* **then**
   | | $v_{\text{free}} \to v_1$ is a pseudo-augmenting path. Pseudo-augment the pseudo-matching $M$. Go to Step 1;
   | **end**
   | **if** $v_1$ *is pseudo-matched to* $z$ **then**
   | | extend the pseudo-matching tree: $S = S \cup \{z\}, T = T \cup \{v_1\}$;
   | | **for** $\mathbf{v_1} \in V_1$ *is matched to* $z$ **do**
   | | | $T = T \cup \mathbf{v_1}$;
   | | **end**
   | **end**
   | Go to Step 2.
   **end**

---

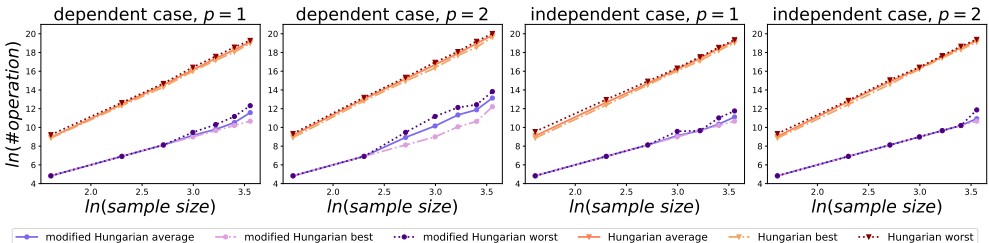

Figure 3: Comparison with the Hungarian algorithm on synthetic data

*where $C^\dagger$ is an $M \times n$ matrix generated by duplicating the $i$th row of $C$ $n_i$ times:*

$$C^\dagger_{tj} = \begin{cases} C_{1j} & 1 \leq t \leq n_1 \\ C_{ij} & n_1 + \cdots + n_{i-1} + 1 \leq t \leq n_1 + \cdots + n_i, 2 \leq i \leq m \end{cases}.$$

Problem (6) belongs to the special type of OT problem (2). We could apply the proposed modified Hungarian algorithm to problem (6) and get the exact solution to problem (4). The resulting computational complexity is $\mathcal{O}(M^2 n)$.

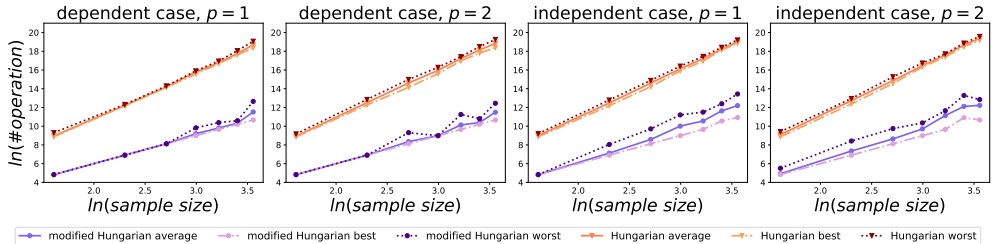

Figure 4: Comparison with the Hungarian algorithm on CIFAR10

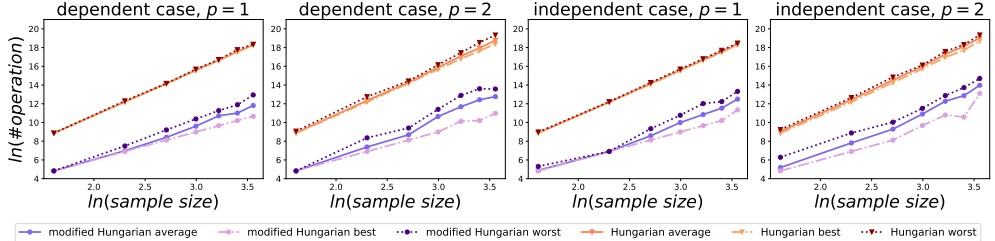

Figure 5: Comparison with the Hungarian algorithm on Wisconsin breast cancer data

### 3.6 Compare with the Hungarian algorithm

We discuss the modification and the improvement of the proposed modified Hungarian algorithm

Our algorithm modifies the Hungarian algorithm to solve a wider class of problems. Hungarian algorithm specializes in the assignment problem which corresponds to the matching in the bipartite graph. The matching is embedded with the 'one-to-one' structure. However, as shown in the solution structure of problem (3), the problems considered in this paper have a 'one-to-many' structure. To deal with the 'one-to-many' structure, pseudo-matching is defined. The proposed algorithm is developed to identify the perfect pseudo-matching while the Hungarian algorithm identifies the perfect traditional matching. In addition, instead of building the augmenting paths in the Hungarian algorithm, we establish the pseudo-augmenting path in the proposed algorithm as shown in Figure 2.

Our algorithm has a lower order of computational complexity. The complexity of converting the problems to the assignment problem and solving them by the Hungarian algorithm only depends on the larger size of the marginals—$\mathcal{O}(m^3)$. To reduce the complexity, we force the pseudo-augmenting paths emanating from $V_2$, which corresponds to the marginal with the smaller size. In this way, the complexity of the direct application of the proposed algorithm depends on the sizes of both marginals—$\mathcal{O}(m^2 n)$. Hence, the proposed modified Hungarian algorithm will outperform, especially when $m \gg n$.

## 4 Application to the independence test using the Wasserstein distance

In this section, we apply the modified Hungarian algorithm to the Wasserstein-distance-based independence test, which originally motivates us to study the special type of OT problem (2).

Suppose that there are $n$ i.i.d. samples $\{(y_1, z_1), \cdots, (y_n, z_n)\}$, where $(y_i, z_i) \sim (Y, Z), Y \sim \nu_1, Z \sim \nu_2$. Recall $\pi$ denotes the joint distribution of $Y, Z$, and one could prove the following equivalence:

$$Y \perp Z \iff \nu_1 \otimes \nu_2 = \pi \iff W(\pi, \nu_1 \otimes \nu_2) = 0,$$

which follows from the fact that the Wasserstein distance is a valid metric between probability measures. Given the empirical data, we utilize the statistic $W(\widehat{\pi}, \widehat{\nu}_1 \otimes \widehat{\nu}_2)$ to test the independence between $Y$ and $Z$,

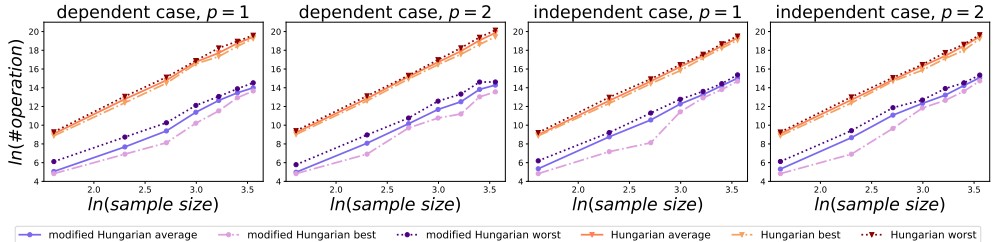

Figure 6: Comparison with the Hungarian algorithm on DOT-benchmark with $512 \times 512$ resolution

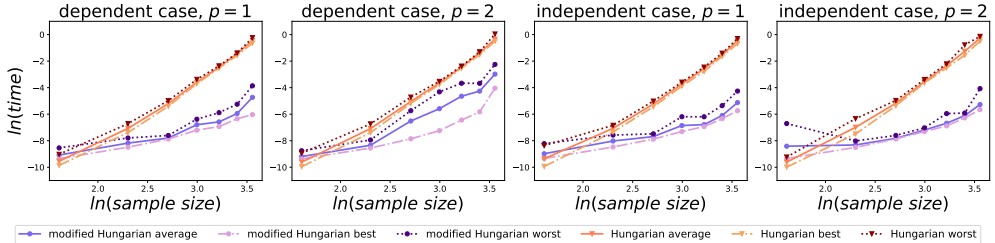

Figure 7: Comparison with the Hungarian algorithm on synthetic data

where $\widehat{\pi}, \widehat{\nu}$ denote the empirical distributions and have the following expressions:

$$
\widehat{\pi} = \begin{pmatrix} \frac{1}{n} & & 0 \\ & \cdots & \\ 0 & & \frac{1}{n} \end{pmatrix}, \quad \widehat{\nu}_1 \otimes \widehat{\nu}_2 = \begin{pmatrix} \frac{1}{n^2} & \cdots & \frac{1}{n^2} \\ \cdots & \cdots & \cdots \\ \frac{1}{n^2} & \cdots & \frac{1}{n^2} \end{pmatrix}.
$$

Plug in the Wasserstein distance formula, the resulting optimization problem is:

$$
\min_{X \in \Pi} \sum_{i,j,k,l=1}^{n} d((y_i, z_j), (y_k, z_l)) X_{ij;kl}, \tag{7}
$$

where

$$
\Pi = \left\{ X_{ij;kl} \geq 0 \,\middle|\, \sum_{k,l=1}^{n} X_{ij;kl} = \frac{1}{n^2}, \sum_{i,j=1}^{n} X_{ij;kl} = \begin{cases} \frac{1}{n} & k = l \\ 0 & k \neq l \end{cases}, \forall i,j,k,l = 1, \cdots, n. \right\}.
$$

It is worth noting that $X_{ij;kl} = 0, k \neq l$. If we let $X_{ij;k}^{\circ} := \sum_{l=1}^{n} X_{ij;kl}$, problem (7) can be simplified as problem (1). Problem (1) belongs to the special type of OT problem, where $m_j = n, \forall j, 1 \leq j \leq n, m = n^2$. Adopting the Hungarian algorithm to problem (1) costs $\mathcal{O}(n^6)$, while adopting the proposed Hungarian algorithm directly costs $\mathcal{O}(n^5)$.

## 5  Application to the one-to-many assignment problem and the many-to-many assignment problem

In this section, we proceed to explain how to apply the modified Hungarian algorithm to solve the one-to-many assignment problem (corresponding to problem (2)) and the many-to-many assignment problem (corresponding to problem (4)). The applications are shown based on two practical examples.

**Example 1 (one-to-many assignment problem):** An assignment problem involving the soccer ball game mentioned by Zhu et al. (2011) is considered here. Suppose a coach is tasked to choose players from a soccer team with $m_1$ players $(a_1, \cdots, a_{m_1})$. There are $d$ roles $(r_1, \cdots, r_d)$. It is assumed that $m_1 > d$. Suppose each player's performance evaluation of each role is known. The overall performance evaluation of the team is the sum of each selected player's performance evaluation of its assigned role. The optimal strategy is to

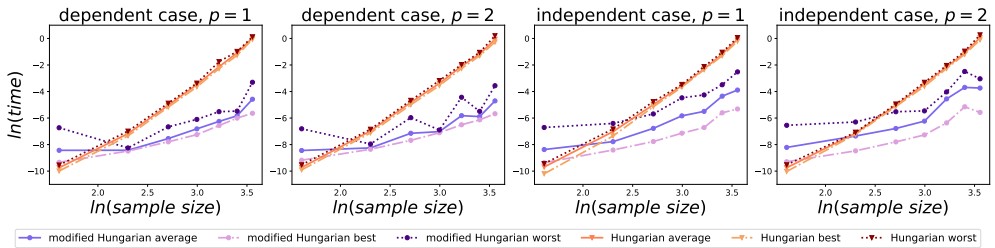

Figure 8: Comparison with the Hungarian algorithm on CIFAR10

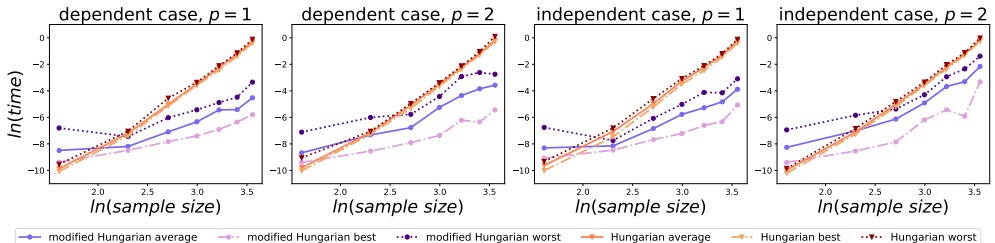

Figure 9: Comparison with the Hungarian algorithm on Wisconsin breast cancer data

maximize the overall performance evaluation of the team. The coach should solve the following optimization problem:

$$\max_{A \in \mathcal{A}} \sum_{i=1}^{m_1} \sum_{j=1}^{d} A_{ij} P_{ij}, \quad \mathcal{A} = \left\{ A_{ij} = \{0,1\} \middle| \sum_{j=1}^{d} A_{ij} \leq 1, \sum_{i=1}^{m_1} A_{ij} = r_j, \forall i = 1, \cdots, m_1; j = 1, \cdots, d \right\},$$

where $P_{ij} \geq 0$ denotes player $a_i$'s performance evaluation of role $j$, $A_{ij} = 1$ means player $a_i$ is selected as role $j$ while $A_{ij} = 0$ means the player is not selected as role $j$.

If $m_1 = \sum_{j=1}^{d} r_j$, the optimization problem above belongs to the special type of OT problem (3), where $n = d, m = \sum_{j=1}^{d} r_j$. The modified Hungarian algorithm could be applied to find the optimal strategy, and the resulting computational complexity is $\mathcal{O}(dm_1^2)$.

If $m_1 > \sum_{j=1}^{d} r_j$, we can't apply the modified Hungarian algorithm directly. To make the problem tractable, we create one more role, and each player's performance evaluation of this role is 0. Players who are not selected are 'assigned' to this role by default. In this scenario, our goal is to solve the following optimization problem:

$$\max_{A^\dagger \in \mathcal{A}^\dagger} \sum_{i=1}^{m_1} \sum_{j=1}^{d+1} A_{ij}^\dagger P_{ij}^\dagger, \quad \mathcal{A}^\dagger = \left\{ A_{ij}^\dagger = \{0,1\} \middle| \sum_{j=1}^{d+1} A_{ij}^\dagger = 1, \sum_{i=1}^{m_1} A_{ij}^\dagger = \begin{cases} r_j & 1 \leq j \leq d \\ m_1 - \sum_{j=1}^{4} r_j & j = d+1 \end{cases} \right\},$$

where we append $P$ by adding one more column of zeros to get $P^\dagger$. It belongs to the special type of OT problem, where $n = d + 1, m = m_1$. Then, we could apply the modified Hungarian algorithm to solve the problem, and the resulting computational complexity is $\mathcal{O}((d+1)m_1^2)$.

Note that the computational order of applying the algorithm developed by Zhu et al. (2011) is $\mathcal{O}(m_1^3)$, which is worse than the proposed modified Hungarian algorithm.

**Example 2 (many-to-many assignment problem):** The following example is an agent-task assignment problem mentioned by Zhu et al. (2016). Assume there are $m_2$ tasks $(t_1, \cdots, t_{m_2})$ and $n_1$ agents $(a_1, \cdots, a_{n_1})$ in total. It is assumed that $n_1 < m_2$. Each task should be undertaken by many agents, and each agent can perform many tasks. To be more specific, task $t_i$ must be assigned to $l_i$ agents, agent $a_j$ can perform at most

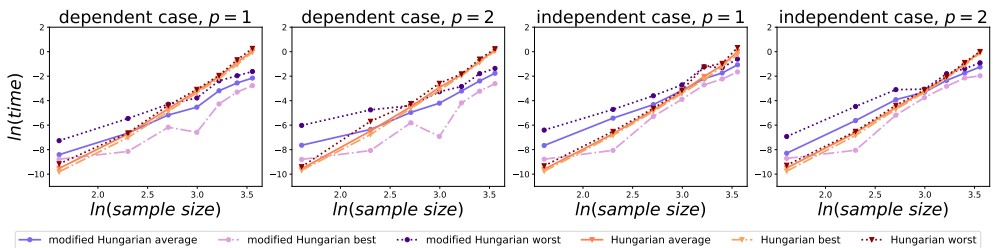

Figure 10: Comparison with the Hungarian algorithm on DOT-benchmark with $512 \times 512$ resolution

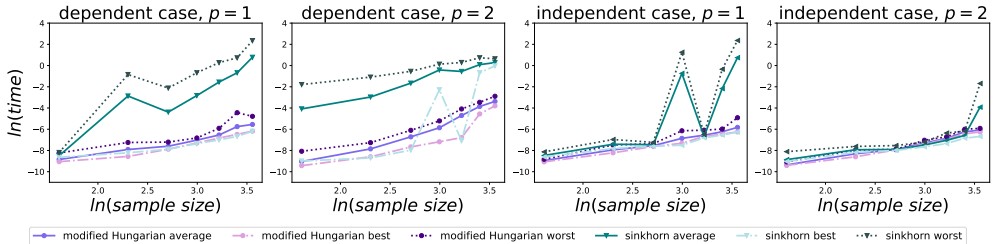

Figure 11: Comparison with the Sinkhorn algorithm on synthetic data

$s_j$ tasks. Suppose the performance evaluation of each agent performing each task is known. The optimal assignment plan is to maximize the overall performance. The resulting optimization problem is as follows:

$$\max_{A' \in \mathcal{A}'} \sum_{i=1}^{m_2} \sum_{j=1}^{n_1} A'_{ij} P'_{ij}, \quad \mathcal{A}' = \left\{ A'_{ij} = \{0,1\} \middle| \sum_{j=1}^{n_1} A'_{ij} = l_i, \sum_{i=1}^{m_2} A'_{ij} \leq s_j, \forall i = 1, \cdots, m_2; j = 1, \cdots, n_1 \right\},$$

where $P'_{ij} \geq 0$ denotes agent $a_j$'s performance evaluation on task $t_i$, $A'_{ij} = 1$ means that agent $a_j$ is assigned to perform task $t_i$ while $A'_{ij} = 0$ means that agent $a_j$ is not assigned to perform task $t_i$.

If $\sum_{i=1}^{m_2} l_i = \sum_{j=1}^{n_1} s_j$, the optimization problem follows the formulation of the problem (4), where $n = n_1, M = \sum_{j=1}^{n_1} s_j$. We could apply the modified Hungarian algorithm to find the optimal assignment plan, and the resulting computational complexity is $\mathcal{O}(n_1 (\sum_{j=1}^{n_1} s_j)^2)$.

If $\sum_{i=1}^{m_2} l_i < \sum_{j=1}^{n_1} s_j$, we create one more task which must be performed by $(\sum_{j=1}^{n_1} s_j - \sum_{i=1}^{m_2} l_i)$ agents, and each agent's performance of this new task equals 0. This reformulation promises that each agent performs the maximum amount of tasks. Accordingly, we need to solve the following optimization problem:

$$\max_{A^{\ddagger} \in \mathcal{A}^{\ddagger}} \sum_{i=1}^{m_2+1} \sum_{j=1}^{n_1} A^{\ddagger}_{ij} P^{\ddagger}_{ij}, \quad \mathcal{A}^{\ddagger} = \left\{ A^{\ddagger}_{ij} = \{0,1\} \middle| \sum_{j=1}^{n_1} A^{\ddagger}_{ij} = \begin{cases} l_i & 1 \leq i \leq m_2 \\ \sum_{j=1}^{n_1} s_j - \sum_{i=1}^{m_2} l_i & j = m_2 + 1 \end{cases}, \sum_{i=1}^{m_2+1} A^{\ddagger}_{ij} = s_j, \right\},$$

where we append $P'$ by adding one more row of zeros to get $P^{\ddagger}$. It follows the formulation of problem (4), where $n = n_1, M = \sum_{j=1}^{n_1} s_j$. We could adopt the method introduced in Section 3.5, and the resulting computation complexity is $\mathcal{O}(n_1 (\sum_{j=1}^{n_1} s_j)^2)$.

Note that the computational order of applying the algorithm developed by Zhu et al. (2016) is $\mathcal{O}((\sum_{j=1}^{n_1} s_j)^3)$, which has a higher computational burden than our proposed method.

## 6 Numerical experiments

In this section, we carry out experiments on the Wasserstein independence test problem on a synthetic dataset, CIFAR10[1] (Krizhevsky et al., 2009), Wisconsin breast cancer dataset[2] (Dua & Graff, 2017) and

---

[1] https://www.cs.toronto.edu/~kriz/cifar.html

[2] https://archive.ics.uci.edu/ml/datasets/Breast+Cancer+Wisconsin+(Diagnostic)

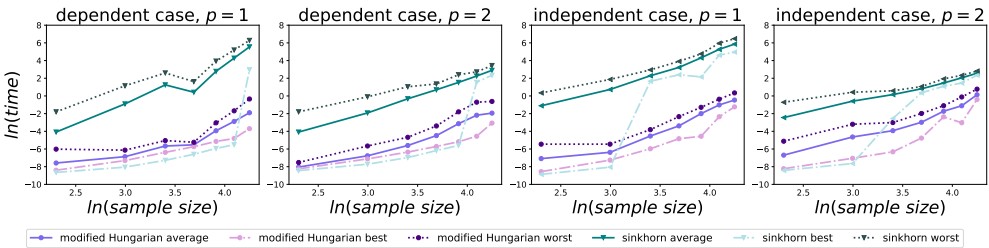

Figure 12: Comparison with the Sinkhorn algorithm on CIFAR10

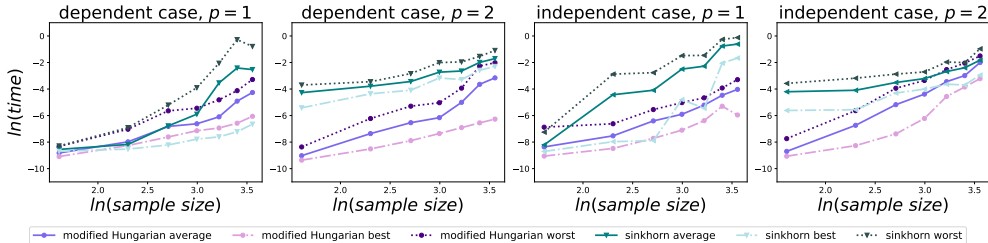

Figure 13: Comparison with the Sinkhorn algorithm on Wisconsin breast cancer data

DOT-benmark (Schrieber et al., 2016). We compare the proposed modified Hungarian algorithm with the Hungarian algorithm, the Sinkhorn algorithm, and the network simplex algorithm. The numerical results show the favorability of applying the proposed algorithm over the Hungarian algorithm, the Sinkhorn algorithm, and the network simplex algorithm.

## 6.1 Experiment setting

Our algorithm is adaptive to any metric. The foregoing experiments are based on the $l_p$ norm-based metric: $d((x_i, y_j), (x_k, y_l)) = \|x_i - x_k\|_p + \|y_j - y_l\|_p$. More specifically, we examine how the modified Hungarian algorithm, the Hungarian algorithm and the Sinkhorn algorithm perform when $p = 1$ and $p = 2$. We create one dependent case and one independent case with different sample sizes for each dataset and run the algorithms on each case 10 times. We plot the worst, best and average number of numerical operations and/or running time for each case.

**Synthetic data**: Suppose that there are independent variables $X \sim N(5\mathbf{1}_{10}, 30I_{10})$, where $\mathbf{1}_{10}$ is a 10-dimensional vector with all ones and $I_{10}$ is the identity matrix; and $Y = (Y_1, .., Y_{25})^T$, where $Y_i$'s are independent and follow Unif(10, 20). We calculate the empirical Wasserstein distance in (1) *independent case*: between $X$ and $Y$; (2) *dependent case*: between $X$ and $Z$ (where $Z = X_1 + Y_1$, $X_1$ is the first 5 coordinates of $X$, $Y_1$ is the first 5 coordinates of $Y$).

**Breast cancer data:** There are 569 instances, and each instance possesses 30 features. Each instance is a 30-dimensional vector, and we rescale the components to $[0, 1]$. There are two classes of instances: benign and malignant. Let $X \in \mathbb{R}^{30}$ be the distribution generated uniformly from the benign class, and $Y \in \mathbb{R}^{30}$ be the distribution generated uniformly from the malignant class. We calculate empirical Wasserstein distance in (1) *independent case*: between $X_1$ and $Y_2$ (where $X_1$ is the first 5 coordinates of $X$, $Y_2$ the last 25 coordinates of $Y$); (2) *dependent case*: between $X$ and $Z$ (where $Z = X_1 * Y_1$, $X_1$ is the first 5 coordinates of $X$, $Y_1$ is the first 5 coordinates of $Y$, $*$ means the coordinate-wise product).

**CIFAR10**: Each image in CIFAR10 contains $32 \times 32$ pixels, and each pixel is composed of 3 color channels. Each image is essentially a 3072-dimensional vector. Then, we rescale the vector components to $[0, 1]$. Suppose $X \in \mathbb{R}^{3072}$ is the distribution generated uniformly from the images of classes: airplane, automobile, bird, cat, and deer; $Y \in \mathbb{R}^{3072}$ is the distribution generated uniformly from the images of other five classes. We calculate the empirical Wasserstein distance in (1) *independent case*: between $X$ and $Y_1$ (where $Y_1$ is the

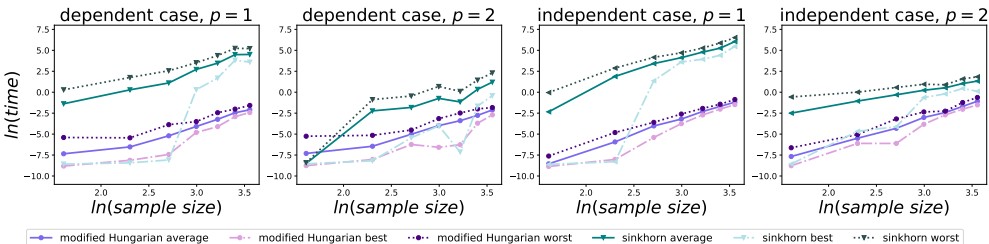

Figure 14: Comparison with the Sinkhorn algorithm on DOT-benchmark with $512 \times 512$ resolution

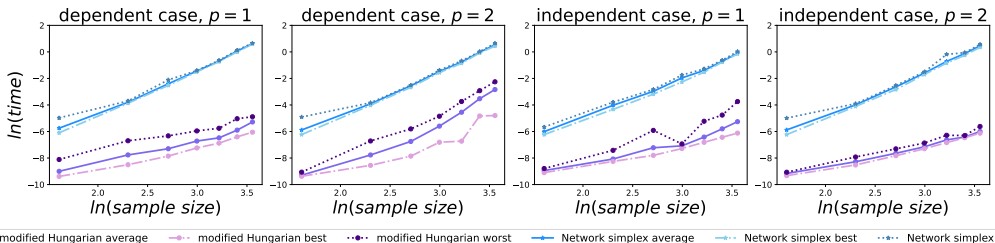

Figure 15: Comparison with the network simplex algorithm on synthetic data

first 1536 coordinates of $Y$); (2) *dependent case*: between $X$ and $Z$ (where $Z = X_2/2 + Y_1/2$, $X_2$ is the last 1536 coordinates of $X$, $Y_1$ is the first 1536 coordinates of $Y$).

DOT-benchmark contains images with different resolutions from 10 classes. Each image is essentially a $r \times r$-dimensional vector, where $r = 32, 64, 128, 256, 512$. Then, we rescale the vector components to $[0, 1]$. Suppose $X \in \mathbb{R}^{r \times r}$ is the distribution generated uniformly from the images of classes: GRFrough, RFmoderate, CauchyDensity, MicroscopyImages, Shapes; $Y \in \mathbb{R}^{r \times r}$ is the distribution generated uniformly from the images of other five classes. We calculate the empirical Wasserstein distance in (1) *independent case*: between $X$ and $Y$ ; (2) *dependent case*: between $X$ and $Z$ (where $Z = X/2 + Y/2$). To save space, we relegate the numerical results when $r = 32, 64, 128, 256$ to the Appendix.

## 6.2   Comparison with the Hungarian algorithm

We compare the modified Hungarian algorithm with the classic Hungarian algorithm. The results in terms of numerical operations are presented in Figure 3, 4, 5, 6. The figures illustrate that the proposed algorithm gains a factor $n$ in computational complexity when solving the proposed special type of OT problem. To be more specific, notice that the slope of ln(number of numerical operations) over ln(sample size) indicates the order of the associated algorithm, and the slope of our algorithm is around 5 while the slope of the Hungarian algorithm is around 6. This observation implies that the order of applying our algorithm is $\mathcal{O}(n^5)$ while the order of applying the Hungarian algorithm is $\mathcal{O}(n^6)$. Such observations are consistent with our theoretical results.

The results in terms of running time are presented in Figure 7, 8, 9,10. One may observe that for almost all instances, especially for larger sample size $n$, the modified Hungarian algorithm is faster than the Hungarian algorithm. It indicates the practical improvement of the modified Hungarian algorithm over the classic Hungarian algorithm.

## 6.3   Comparison with the Sinkhorn algorithm

We compare the modified Hungarian algorithm with the Sinkhorn algorithm. Among the state-of-the-art approximation solvers for OT problems, the first-order approximation algorithms (Dvurechensky et al., 2018; Lin et al., 2019b; Guo et al., 2020) are mainly employed to solve the balanced case ($m = n$). The Sinkhorn algorithm could deal with the unbalanced scenario ($m \neq n$) and is widely used in all kinds of OT-related

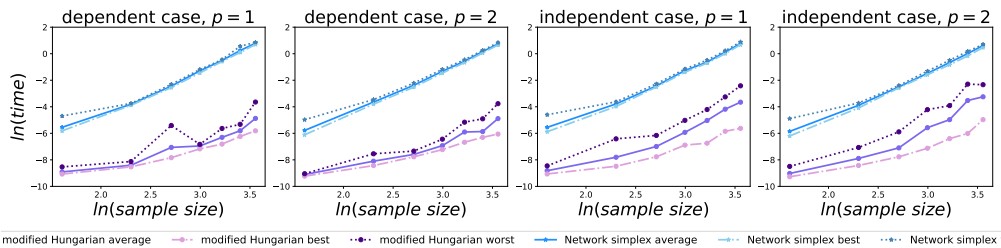

Figure 16: Comparison with the network simplex algorithm on CIFAR10

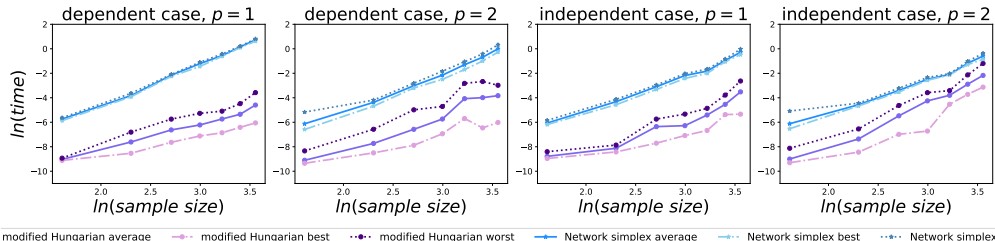

Figure 17: Comparison with the network simplex algorithm on Wisconsin breast cancer data

models. Therefore, we choose the Sinkhorn algorithm as the baseline and then investigate the performance of the Sinkhorn algorithm in the Wasserstein-distance-based independence test problem. When we implement the Sinkhorn algorithm, we set the regularization parameter as 0.1 and the accuracy as 0.0001.

The results in terms of running time are presented in Figure 11, 12, 13,14. We relegate the results in terms of numerical operations to the Appendix. For most of the scenarios, the average running time of the modified Hungarian algorithm is less than the Sinkhorn algorithm. Moreover, the performance of the modified Hungarian algorithm has a lower variance than the Sinkhorn algorithm. The results demonstrate that our proposed modified Hungarian algorithm should be chosen if one is interested in obtaining solutions with a high accuracy.

### 6.4 Comparison with the network simplex algorithm

We compare the modified Hungarian algorithm with the network simplex algorithm. The results are presented in Figure 15, 16, 17, 18. We run the network simplex from networkX library in Python. Considering the package requires integer-valued input, we round the costs to the nearest integers from the below. According to the experimental results, we could conclude that the modified Hungarian algorithm is superior to the network simplex algorithm.

## 7 Discussion

A modified Hungarian algorithm is developed to efficiently solve a wide range of OT problems. Theoretical analysis and numerical experiments demonstrate that the proposed algorithm compares favorably with the Hungarian algorithm and the Sinkhorn algorithm. In addition to the computational aspects, broad applications are explored, including the Wasserstein-distance-based independence test, the one-to-many assignment problem and the many-to-many assignment problem. The many-to-many assignment problem closely relates to practical problems involving service assignment problems (Ng et al., 2008), sensor networks (Bhardwaj & Chandrakasan, 2002), and access control (Ahn & Hu, 2007). Future work along this line is to apply the proposed algorithm to problems involving engineering and control. Also, there is some possibility of applying the proposed algorithm to some unsupervised learning problems. For example, the clustering problem could be formulated as an OT problem (Genevay et al., 2019). Assume that there are $n$ clusters and $m$ samples in total, and each cluster has $m_j$ samples. If we want to identify the cluster assignment to minimize the

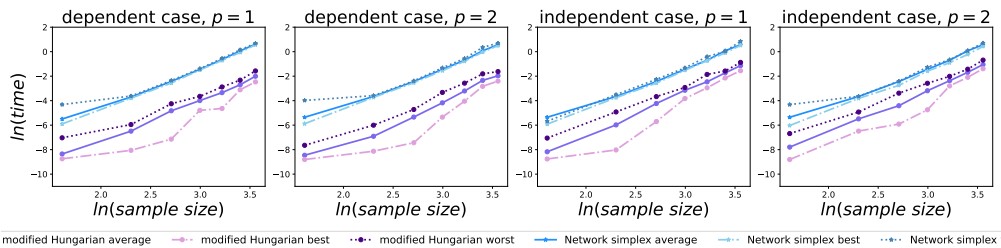

Figure 18: Comparison with the network simplex algorithm on DOT-benmark with $512 \times 512$ resolution

'distance' between cluster 'centers' and the associated assigned samples, we are solving the special type of OT problem in this paper. The future work along this line may be to find a scheme to determine 'distance' and 'centers' to promise desirable model performances.

## Acknowledgement

The authors would like to thank the Action Editor and anonymous reviewers for their detailed and constructive comments, which enhanced the quality and presentation of the manuscript.

This project is partially supported by the Transdisciplinary Research Institute for Advancing Data Science (TRIAD), https://research.gatech.edu/data/triad, which is a part of the TRIPODS program at NSF and locates at Georgia Tech, enabled by the NSF grant CCF-1740776. The authors are also partially sponsored by NSF grants 2015363.

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

# A  Appendix

## A.1  Proof of Proposition 1

*Proof.* We first consider the following two optimization problems (8), (9):

$$\min_{X^1 \in \mathcal{U}^1} \sum_{i=1}^{m} \sum_{j=1}^{m} X_{ij}^1 C_{ij}^{\ddagger}, \quad \mathcal{U}^1 = \left\{ X_{ij}^1 \geq 0 \middle| \sum_{j=1}^{m} X_{ij}^1 = \frac{1}{m}, \sum_{i=1}^{m} X_{ij}^1 = \frac{1}{m}, \forall i,j = 1, \cdots, m \right\}. \tag{8}$$

where $C^{\ddagger}$ is an $m \times m$ matrix generated by duplicating the $j$th column of $C$ $m_j$ times:

$$C_{it}^{\ddagger} = \begin{cases} C_{i1} & 1 \leq t \leq m_1, \\ C_{ij} & m_1 + \cdots + m_{j-1} + 1 \leq t \leq m_1 + \cdots + m_j, 2 \leq j \leq n \end{cases}.$$

$$\min_{X^{\ddagger} \in \mathcal{U}^{\ddagger}} \sum_{i=1}^{m} \sum_{j=1}^{m} \frac{1}{m} X_{ij}^{\ddagger} C_{ij}^{\ddagger}, \quad \mathcal{U}^{\ddagger} = \left\{ X_{ij}^{\ddagger} = \{0,1\} \middle| \sum_{j=1}^{m} X_{ij}^{\ddagger} = 1, \sum_{i=1}^{m} X_{ij}^{\ddagger} = 1, \forall i,j = 1, \cdots, m \right\}, \tag{9}$$

Then, we denote the objective functions of problems (2), (3), (8) and (9) by $f'(X')$, $f(X)$, $f^1(X^1)$, and $f^{\ddagger}(X^{\ddagger})$, respectively.

Firstly, we prove (2) $\iff$ (8).

On one hand, for any $X^1 \in \mathcal{U}^1$, if we let

$$X'_{ij} = \sum_{t=1}^{m_1} X^1_{it}, \quad j = 1,$$

$$X'_{ij} = \sum_{t=m_1+\cdots+m_{j-1}+1}^{m_1+\cdots+m_j} X^1_{it}, \quad 2 \le j \le n,$$

then we have

$$X'_{ij} \ge 0,$$

$$\sum_{j=1}^{n} X'_{ij} = \sum_{t=1}^{m_1} X^1_{it} + \sum_{t=m_1+\cdots+m_{j-1}+1}^{m_1+\cdots+m_j} X^1_{it} = \sum_{t=1}^{m} X^1_{it} = \frac{1}{m},$$

$$\sum_{i=1}^{m} X'_{ij} = \sum_{i=1}^{m}\sum_{t=1}^{m_1} X^1_{it} = \frac{m_1}{m}, \quad j = 1,$$

$$\sum_{i=1}^{m} X'_{ij} = \sum_{i=1}^{m} \sum_{t=m_1+\cdots+m_{j-1}+1}^{m_1+\cdots+m_j} X^1_{it} = \frac{m_j}{m}, \quad 2 \le j \le n.$$

Thus, $X' \in \mathcal{U}'$.

For the objective functions, we have the following:

$$f'(X') = \sum_{i=1}^{m}\sum_{j=1}^{n} X'_{ij} C_{ij} = \sum_{i=1}^{m}\left( \sum_{t=1}^{m_1} X^1_{it} C^1_{it} + \sum_{t=m_1+\cdots+m_{j-1}+1}^{m_1+\cdots+m_j} X^1_{it} C^1_{it} \right) = \sum_{i=1}^{m}\sum_{t=1}^{m} X^1_{it} C^1_{it} = f^1(X^1).$$

On the other hand, for any $X' \in \mathcal{U}$, if we let

$$X^1_{it} = \begin{cases} X'_{i1}/m_1 & 1 \le t \le m_1 \\ X'_{ij}/m_j & m_1+\cdots+m_{j-1}+1 \le t \le m_1+\cdots+m_j, 2 \le j \le n, \end{cases}$$

then we have

$$X^1_{it} \ge 0,$$

$$\sum_{t=1}^{m} X^1_{it} = \sum_{j=1}^{n} \frac{X'_{ij}}{m_j} m_j = \sum_{j=1}^{n} X'_{ij} = \frac{1}{m},$$

$$\sum_{i=1}^{n} X^1_{it} = \sum_{i=1}^{n} \frac{X'_{ij}}{m_j} = \frac{1}{m_j} \sum_{i=1}^{n} X'_{ij} = \frac{1}{m}.$$

Thus, $X^1 \in \mathcal{U}^1$.

For the objective functions, we have the following:

$$f^1(X^1) = \sum_{i=1}^{m}\sum_{t=1}^{m} X^1_{it} C^1_{it} = \sum_{i=1}^{m}\sum_{j=1}^{n} \frac{X'_{ij}}{m_j} C_{it} m_j = f'(X').$$

Hence, (2) $\iff$ (8).

By Birkhoff's theorem, we know (8) $\iff$ (9). Therefore, we have (2) $\iff$ (9).

Similarly, for any $X^\ddagger \in \mathcal{U}^\ddagger$, if we let

$$X_{ij} = \sum_{t=1}^{m_1} X_{it}^\ddagger, \quad j = 1,$$

$$X_{ij} = \sum_{t=m_1+\cdots+m_{j-1}+1}^{m_1+\cdots+m_j} X_{it}^\ddagger, \quad 2 \le j \le n,$$

then we have $X \in \mathcal{U}$ and $f^\ddagger(X^\ddagger) = f(X)$.

For any $X \in \mathcal{U}$, if we let

$$X_{it}^\ddagger = \begin{cases} X_{i1}/m_1, & 1 \le t \le m_1 \\ X_{ij}/m_j & m_1 + \cdots + m_{j-1} + 1 \le t \le m_1 + \cdots + m_j, 2 \le j \le n \end{cases}$$

then we have $X^\ddagger \in \mathcal{U}^\ddagger$ and $f^\ddagger(X^\ddagger) = f(X)$.

Therefore, (3) $\iff$ (9).

In conclusion, we have (3) $\iff$ (9) $\iff$ (2). $\qquad\square$

### A.2 Proof of Theorem 1

*Proof.* Denote the edge $e \in E$ by $e = (e_{v_1}, e_{v_2})$. Let $PM'$ be any perfect pseudo-matching in $G$ (not necessarily in the equality graph $E_l$). And $v_1^i, i = 1, \cdots, m$; $v_2^j, j = 1, \cdots, n$ are nodes from $V_1$ and $V_2$, respectively. Since $v_1^i \in V_1$ is covered exactly once by $PM'$, and $v_2^j \in V_2$ is covered exactly $m_j$ times by $PM'$, we have

$$w(PM') = \sum_{e \in PM'} w(e) \le \sum_{e \in PM'} (l(e_{v_1}) + l(e_{v_2})) = \sum_{i=1}^m l(v_1^i) + \sum_{j=1}^n m_j l(v_2^j),$$

where the first inequality comes from the definition of feasible labeling.

Thus, $\sum_{i=1}^m l(v_1^i) + \sum_{j=1}^n m_j l(v_2^j)$ is the upper bound of the weight of any perfect pseudo-matching. Then let $PM$ be a perfect pseudo-matching in the equality graph $E_l$, we have

$$w(PM) = \sum_{e \in PM} w(e) = \sum_{i=1}^m l(v_1^i) + \sum_{j=1}^n m_j l(v_2^j).$$

Hence $w(PM') \le w(PM)$, and $PM$ is the maximum weighted pseudo-matching. $\qquad\square$

### A.3 Proof of Proposition 2

*Proof.* We denote the objective functions of problems (4) and (6) by $g^*(X^*)$, and $g^\dagger(X^\dagger)$, respectively.

On one hand, for any $X^\dagger \in \mathcal{U}^\dagger$, if we let

$$X_{ij}^* = \sum_{t=1}^{n_1} X_{tj}^\dagger, \quad i = 1,$$

$$X_{ij}^* = \sum_{t=n_1+\cdots+n_{i-1}+1}^{n_1+\cdots+n_i} X_{tj}^\dagger, \quad 2 \le i \le m,$$

then we have

$$X_{ij}^* \ge 0,$$

$$\sum_{i=1}^{m} X_{ij}^* = \sum_{t=1}^{n_1} X_{tj}^\dagger + \sum_{t=n_1+\cdots+n_{i-1}+1}^{n_1+\cdots+n_i} X_{tj}^\dagger = \sum_{t=1}^{M} X_{tj}^\dagger = \frac{m_j}{M},$$

$$\sum_{j=1}^{n} X_{ij}^* = \sum_{j=1}^{n}\sum_{t=1}^{n_1} X_{tj}^\dagger = \frac{n_1}{M}, \quad i = 1,$$

$$\sum_{i=1}^{m} X_{ij}^* = \sum_{i=1}^{m}\sum_{t=n_1+\cdots+n_{i-1}+1}^{n_1+\cdots+n_i} X_{it}^\dagger = \frac{n_i}{m}, \quad 2 \le i \le m.$$

Thus, $X^* \in \mathcal{U}^*$.

For the objective function, we have the following:

$$g^*(X^*) = \sum_{i=1}^{M}\sum_{j=1}^{n} X_{ij}^* C_{ij} = \sum_{j=1}^{n}\left(\sum_{t=1}^{n_1} X_{tj}^\dagger C_{tj} + \sum_{t=n_1+\cdots+n_{i-1}+1}^{n_1+\cdots+n_i} X_{it}^\dagger C_{it}\right) = \sum_{j=1}^{n}\sum_{t=1}^{M} X_{it}^\dagger C_{it}^\dagger = g^\dagger(X^\dagger).$$

On the other hand, for any $X^* \in \mathcal{U}^*$, if we let

$$X_{tj}^\dagger = \begin{cases} X_{1j}^*/n_1, & 1 \le t \le n_1 \\ X_{ij}^*/n_i & n_1+\cdots+n_{i-1}+1 \le t \le n_1+\cdots+n_i, 2 \le i \le m \end{cases}.$$

then we have

$$X_{ij}^\dagger \ge 0,$$

$$\sum_{j=1}^{n} X_{ij}^\dagger = \sum_{j=1}^{n} \frac{X_{ij}^*}{n_i} = \frac{n_i}{M},$$

$$\sum_{i=1}^{M} X_{ij}^\dagger = \sum_{i=1}^{M} \frac{X_{ij}^*}{n_i} n_i = \sum_{i=1}^{m} X_{ij}^* = \frac{m_j}{M}.$$

Thus, $X^\dagger \in \mathcal{U}^\dagger$.

For the objective function, we have the following:

$$g^\dagger(X^\dagger) = \sum_{i=1}^{M}\sum_{j=1}^{n} X_{ij}^\dagger C_{ij}^\dagger = \sum_{j=1}^{n}\sum_{i=1}^{m} \frac{X_{ij}^*}{n_i} C_{it} n_i = g^*(X^*).$$

Hence, (4) $\iff$ (6). $\qquad\square$

### A.4  Additional experiment results

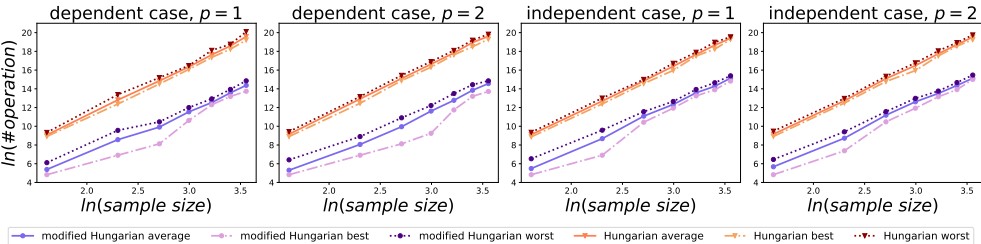

Figure 19: Comparison with the Hungarian algorithm on DOT-benchmark with $32 \times 32$ resolution w.r.t. numerical operations

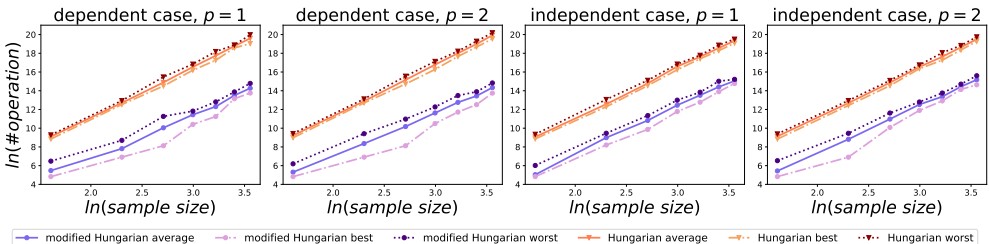

Figure 21: Comparison with the Hungarian algorithm on DOT-benchmark with $128 \times 128$ resolution w.r.t. numerical operations

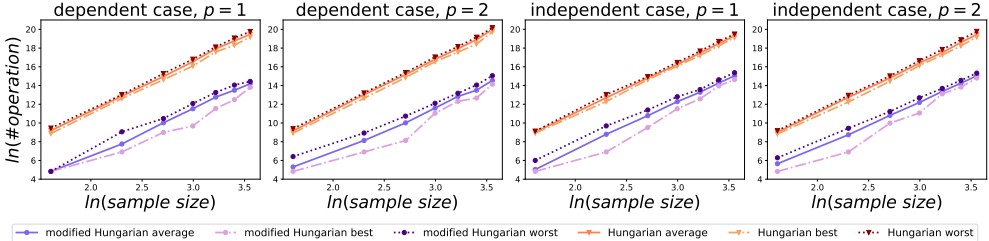

Figure 22: Comparison with the Hungarian algorithm on DOT-benchmark with $256 \times 256$ resolution w.r.t. numerical operations

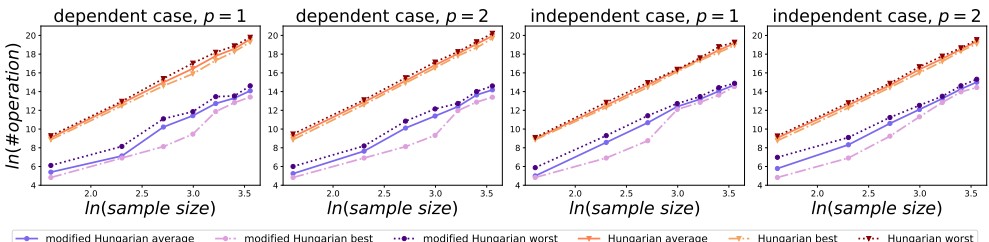

Figure 20: Comparison with the Hungarian algorithm on DOT-benchmark with $64 \times 64$ resolution w.r.t. numerical operations

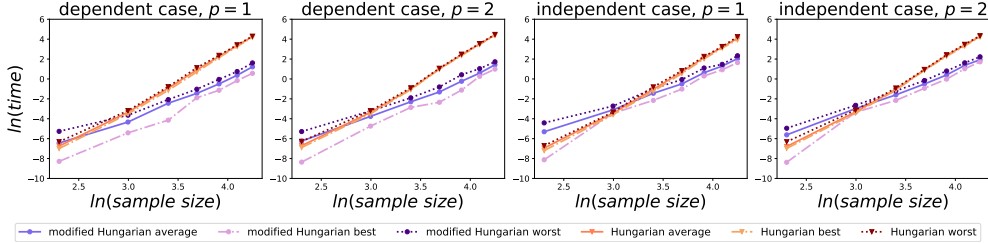

Figure 23: Comparison with the Hungarian algorithm on DOT-benchmark with $32 \times 32$ resolution w.r.t. running time

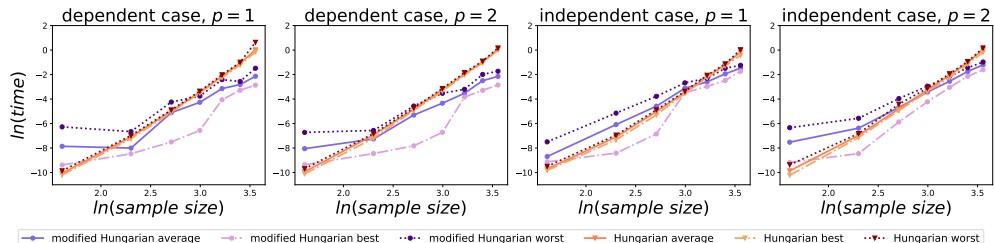

Figure 24: Comparison with the Hungarian algorithm on DOT-benchmark with $64 \times 64$ resolution w.r.t. running time

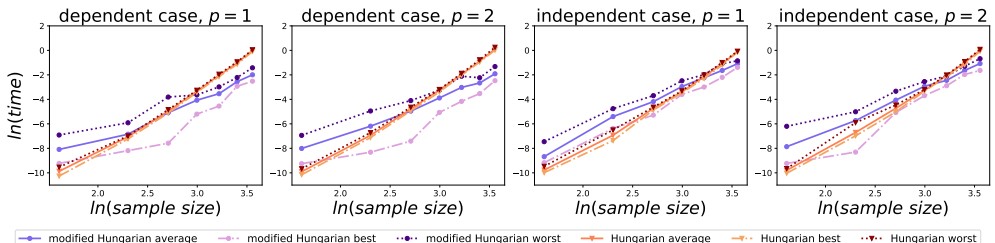

Figure 25: Comparison with the Hungarian algorithm on DOT-benchmark with $128 \times 128$ resolution w.r.t. running time

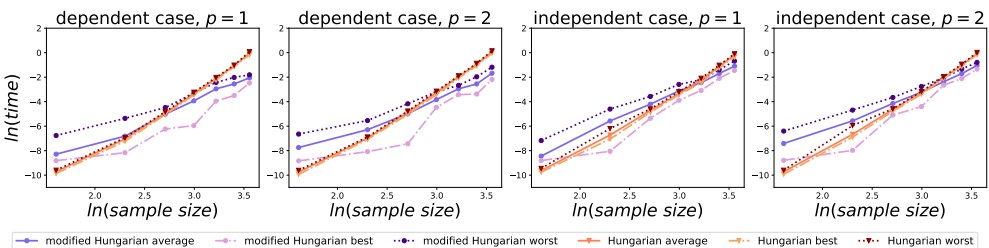

Figure 26: Comparison with the Hungarian algorithm on DOT-benchmark with $256 \times 256$ resolution w.r.t. running time

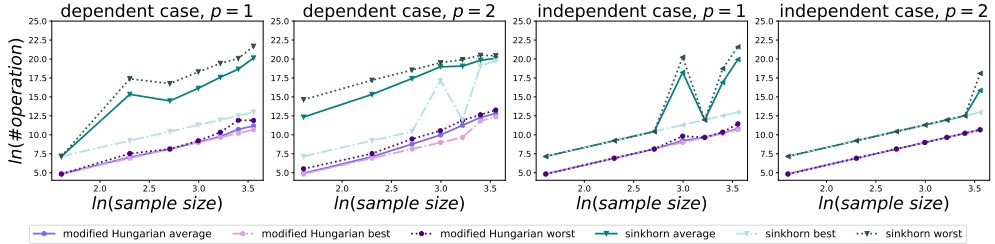

Figure 27: Comparison with the Sinkhorn algorithm on synthetic data w.r.t. numerical operations

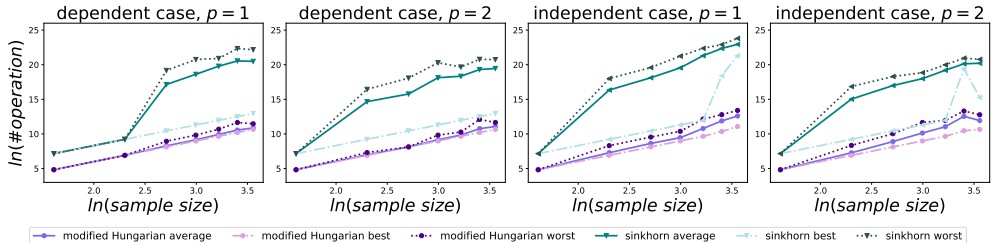

Figure 28: Comparison with the Sinkhorn algorithm on CIFAR10 w.r.t. numerical operations

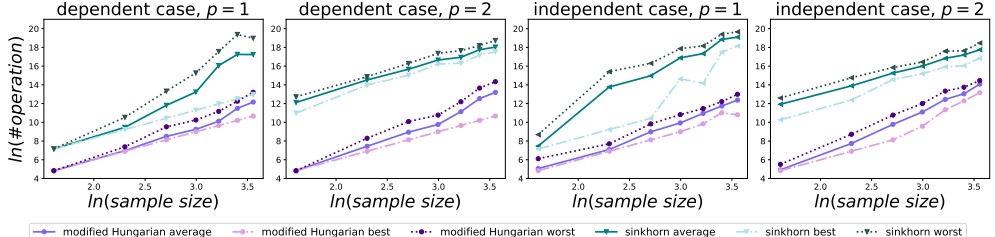

Figure 29: Comparison with the Sinkhorn algorithm on Wisconsin cancer data w.r.t. numerical operations

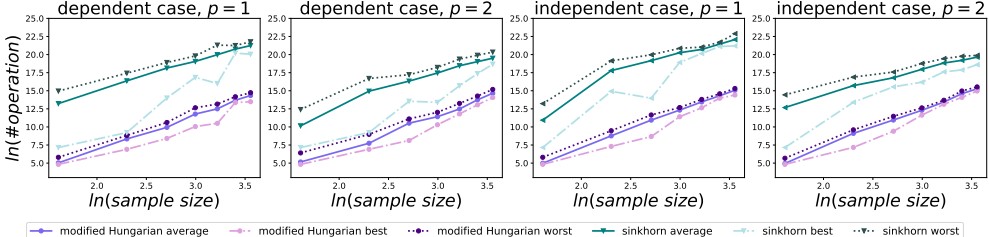

Figure 30: Comparison with the Sinkhorn algorithm on DOT-benchmark with $32 \times 32$ resolution w.r.t. numerical operations

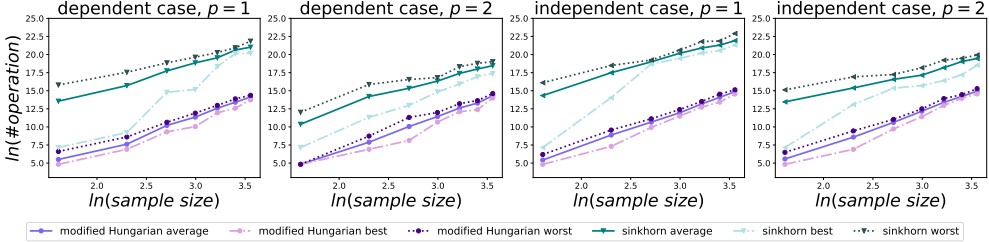

Figure 31: Comparison with the Sinkhorn algorithm on DOT-benchmark with $64 \times 64$ resolution w.r.t. numerical operations

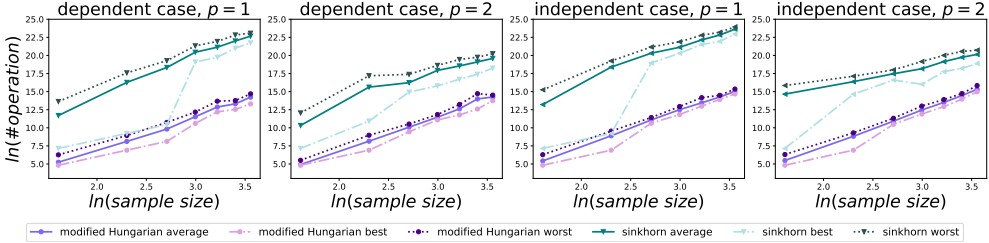

Figure 32: Comparison with the Sinkhorn algorithm on DOT-benchmark with $128 \times 128$ resolution w.r.t. numerical operations

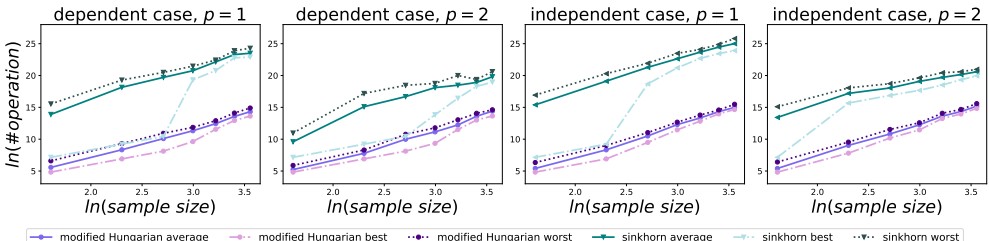

Figure 33: Comparison with the Sinkhorn algorithm on DOT-benchmark with $256 \times 256$ resolution w.r.t. numerical operations

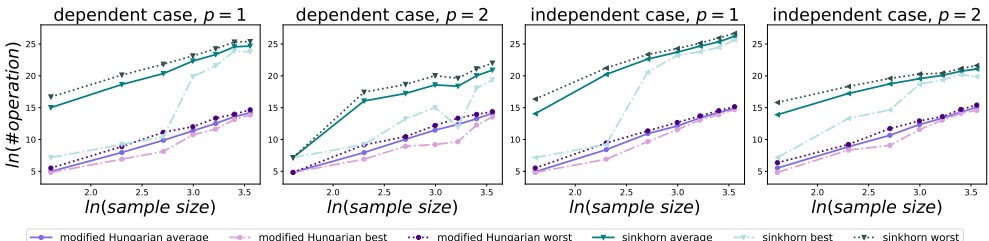

Figure 34: Comparison with the Sinkhorn algorithm on DOT-benchmark with $512 \times 512$ resolution w.r.t. numerical operations

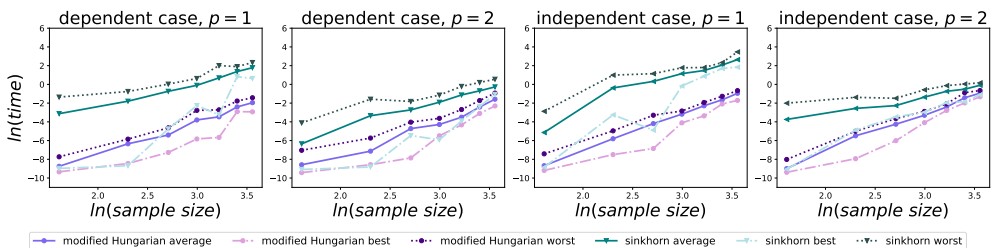

Figure 35: Comparison with the Sinkhorn algorithm on DOT-benchmark with $32 \times 32$ resolution w.r.t. running time

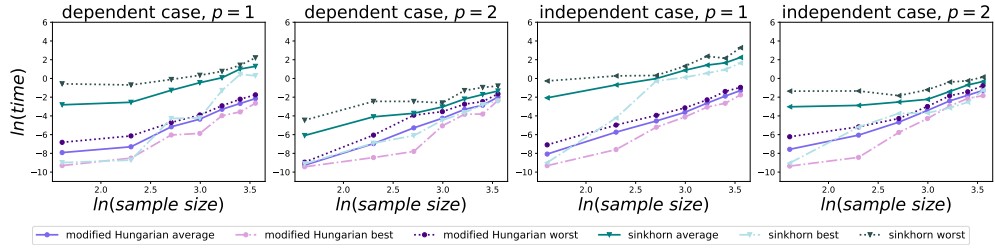

Figure 36: Comparison with the Sinkhorn algorithm on DOT-benchmark with $64 \times 64$ resolution w.r.t. running time

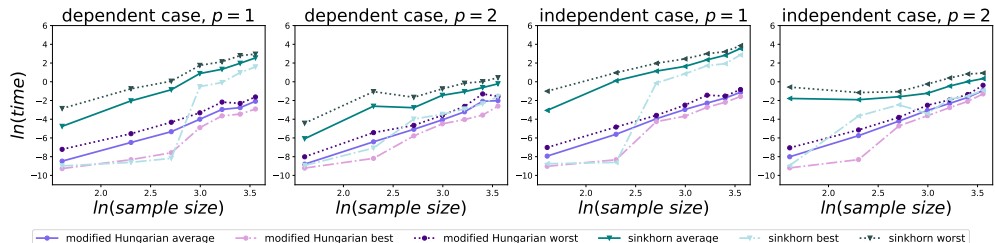

Figure 37: Comparison with the Sinkhorn algorithm on DOT-benchmark with $128 \times 128$ resolution w.r.t. running time

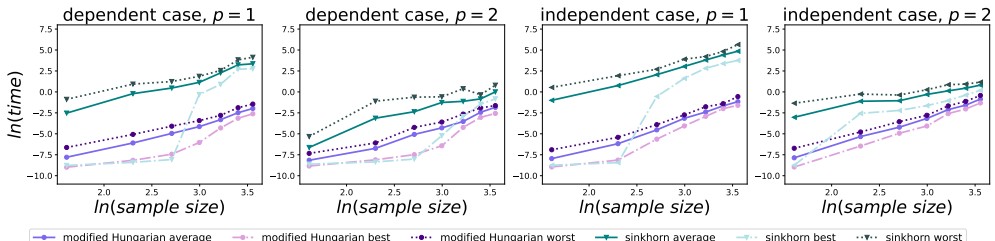

Figure 38: Comparison with the Sinkhorn algorithm on DOT-benchmark with $256 \times 256$ resolution w.r.t. running time

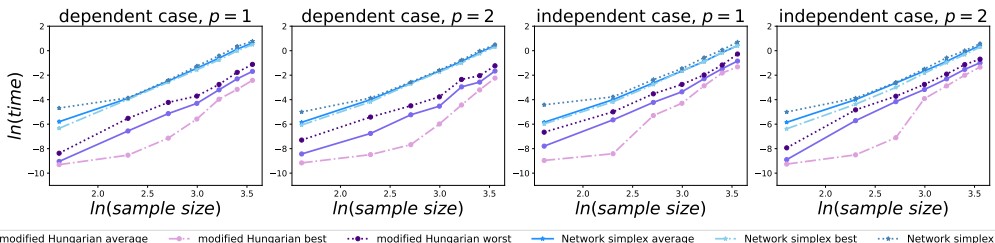

Figure 39: Comparison with the network simplex algorithm on DOT-benchmark with $32 \times 32$ resolution w.r.t. running time

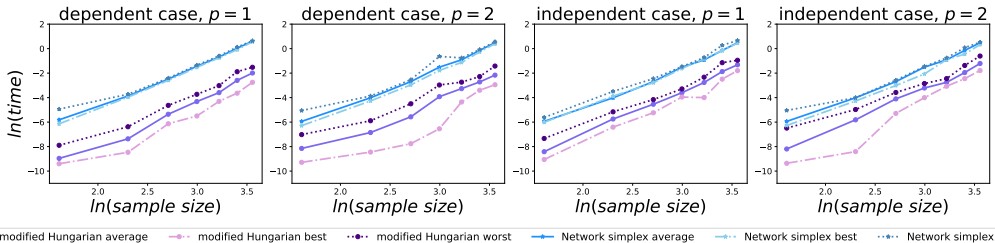

Figure 40: Comparison with the network simplex algorithm on DOT-benchmark with $64 \times 64$ resolution w.r.t. running time

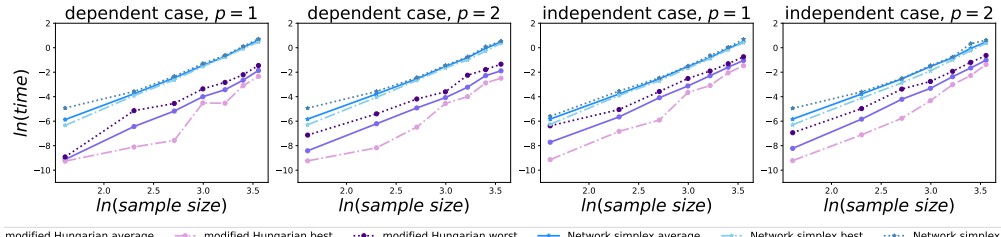

Figure 41: Comparison with the network simplex algorithm on DOT-benchmark with $128 \times 128$ resolution w.r.t. running time

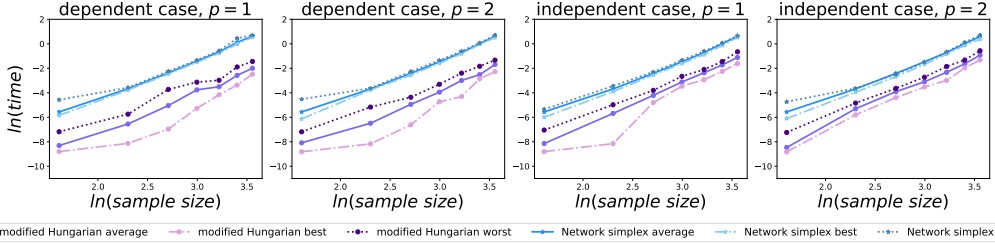

Figure 42: Comparison with the network simplex algorithm on DOT-benchmark with $256 \times 256$ resolution w.r.t. running time

