# OpenReview forum: "Solving a Special Type of Optimal Transport Problem by a Modified Hungarian Algorithm"
_TMLR — Accepted by TMLR_

### Review · Reviewer_F217 · 2022-12-07

**Summary Of Contributions:**

The authors prove computational complexity guarantees for a modified Hungarian algorithm to solve a special type of discrete optimal transport problem. They showcase applications to independence testing and other assignment problems.

**Audience:**

Yes

**Broader Impact Concerns:**

None.

**Claims And Evidence:**

No

**Requested Changes:**

As discussed above, please conduct a more thorough literature search and provide a detailed comparison with the network simplex algorithm, among other algorithms; adjust the claims accordingly.

Also, the phrase “w.l.o.g.” is used incorrectly throughout. To be clear, WLOG means we can focus on a special case, because the general case is reducible to this special case. In this paper, WLOG seems to be used to mean “we impose this assumption as this is the situation that we care about”.

The notation $\gamma(\nu_1, \nu_2)$ is never defined; presumably it refers to the joint distribution of $(Y, Z)$. The notation does not make sense; the joint distribution is not a “function” of the marginal distributions. Moreover it is confusing as it is easily confused for the optimal transport plan with marginals $\nu_1$ and $\nu_2$.


**Strengths And Weaknesses:**

The strength of the work is that the proposed Hungarian algorithm indeed has a good complexity for this problem.

The weaknesses are that the paper is not well-written, and the comparison to the literature is inadequate. Indeed, in the (short) related work section, the proposed approach is only compared to low-accuracy OT solvers, but it seems like a more appropriate comparison is with the combinatorial algorithms described in Peyré et al. (2019) (by the way, the bibliography entry for this reference is incorrect, as it says Peyré, Cuturi, et al., but I am only aware of two authors total), sections 3.5-3.7. In particular, as mentioned at the end of section 3.5, it is already known that the network simplex algorithm only requires $\widetilde O(m^2 n)$ complexity. Without a detailed comparison to prior work, the submission cannot be accepted for publication.

---

> ### Author Response · Authors · 2022-12-14
> **Response to Reviewer F217**
>
> We thank the reviewer for the careful comments, and we reply the comments as following:
> \
> \
> $\bullet$  **A detailed comparison with the network simplex algorithm, among other algorithms**
>
> We make a comparison with the combinatorial algorithms mentioned in Section 3.5-3.7 of [4] as below. *(A more formal presentation could be found in **Related work** in the most recent revised version of our paper.)*
>
> $\ast$ Orlin [3] proposes the first polynomial-time **network simplex algorithm**, and Tarjan [5] further improves Orlin's result. The associated computational complexity of applying the Tarjan's algorithm to the special type of optimal transport problem is $\mathcal{O}(m^2n\log(m)\min\{ \log(m C_{max}),mn\log (m)\})$, where $C_{max}$ denotes the maximum absolute value of the costs if all costs are integers and $\infty$ otherwise.  More specifically, if  the costs are integral, the resulting computational complexity is $\mathcal{O}(m^2n\log(m)\log(m C_{max}))$, which is comparable to the proposed algorithm; if the costs are not integral, the resulting computational complexity is $\mathcal{O}(m^3n^2\log^2(m))$, which is much worse than our algorithm.
>
>
>  $\ast$ The output of the **auction algorithm** is suboptimal (Proposition 3.9 in [4]), while the output of our algorithm is optimal.
>  Furthermore, Theorem 7.16 in [2] and [1] illustrate that finite termination of the auction algorithm with an optimal solution can come from the nature of integer-valued cost coefficients.
>
>
>  $\ast$ Theorem 7.13 in [2] demonstrates that integral cost input is one of the necessary conditions for finite termination of **dual ascent algorithm**.
>
>
>  In conclusion, fast computation or finite termination  with an optimal solution of the aforementioned combinatorial algorithms require *integer*-valued costs, while our modified Hungarian algorithm can handle *real*-valued costs. Although most of the cases in practice may have rational costs, which can be scaled to integer-valued, it is more convenient to use the proposed algorithm since it allows the direct input of any real-valued costs without additional scaling step. Also, looking into problems with real-valued costs is of much theoretical interest.
>
> \
> $\bullet$ **Incorrect reference of [4]; incorrect use of `w.l.o.g.' and the notation $\gamma(\nu_1,\nu_2)$**
>
> We have fixed these issues in the most recent revised version of our paper.
> \
> \
> \
> **References**
>
> [1] D. P. Bertsekas. The auction algorithm: A distributed relaxation method for the assignment
> problem. Annals of operations research, 14(1):105–123, 1988.
>
> [2] D. Bertsimas and J. N. Tsitsiklis. Introduction to linear optimization, volume 6. Athena Scientific
> Belmont, MA, 1997.
>
> [3] J. B. Orlin. A polynomial time primal network simplex algorithm for minimum cost flows. Mathematical Programming, 78(2):109–129, 1997.
>
> [4] G. Peyre and M. Cuturi. Computational optimal transport. Foundations and Trends in Machine Learning, 11(5-6):355–607, 2019.
>
> [5] R. E. Tarjan. Dynamic trees as search trees via euler tours, applied to the network simplex algorithm. Mathematical Programming, 78(2):169–177, 1997.

---

> > ### Comment · Reviewer_F217 · 2023-01-21
> > **Response**
> >
> > Thank you for your response, I am satisfied by the changes.

---

### Review · Reviewer_ayht · 2022-12-21

**Summary Of Contributions:**

This paper shows how the Wasserstein distance-based independence test between two empirical discrete probability measures boils down to the solution of a semi-assignment problem.

The authors claim to have a novel modification of the Hungarian algorithm for computing minimum cost bipartite matching with worst-case complexity of $O(n^5)$, which is better than applying the Hungarian algorithm with a complexity of $O(n^6)$, where n is the number of i.i.d. samples being tested for independence.

The proposed algorithm is implemented in Python and compared with the Sinkhorn-based algorithm on the CIFAR10 and the Wisconsin breast cancer datasets. In the proposed instances, the runtime of their algorithm looks modestly interesting.


**Audience:**

Yes

**Broader Impact Concerns:**

We have no major concerns.

**Claims And Evidence:**

No

**Requested Changes:**

The main request for changes is concerned with better positioning of the presented optimal transport problem in the cost of existing network problems.

The best sequential algorithm to solve the semi-assignment problem is presented in Kennington, J. and Wang, Z., 1992 cited before, which has the same worst-case complexity as the algorithm proposed in this paper. Moreover, that algorithm was further improved in parallel implementations in subsequent papers from other authors.

Given that the algorithm proposed in this paper does not have a better worst-case complexity, the authors should compare their approach with the algorithm in Kennington, J. and Wang, Z., 1992.

If they believe the effort to re-implement their algorithm is too significant, at least they should compare with the solution of the semi-assignment problem via the Network Simplex as implemented in the COIN-OR Lemon Graph library, which in the paper on arxiv by Dong et. all 2020 (that you cite) is clearly one of the best implementation to solve minimum cost network flow problems.

More importantly, you should clearly explain how your algorithm differs from the Kennington, J. and Wang, Z., 1992 algorithm because it does not look too different.

Section 4 does not add too much value to the paper, and it could be merged with the introduction.

Other requests for changes are implicit in the definition of the weakness of the paper (previous section).

Minor comments:
- In problem formulation (1), in the definition of PI^o is unclear the quantification over the constraints (for all ij in … ?)
- In problem (2), are the m_j variable or parameters? The last constraint on the summation of m_j does not look like a constraint but an assumption on the data of the problem.
- Problem (3) is known in the literature as a semi-assignment problem and in general, can be formulated and solved as a minimum cost network flow problem (see the related citation)
- We suggest including the plots of the computational results with the running time in the paper, and postponing in the appendix the results with the number of operations. The number of operations can be a misleading measure of complexity. For instance, the Sinkhorn algorithm is easily parallelizable, while primal-dual algorithms, like the  Hungarian algorithm, are much harder to parallelize and do not get the same speed-up gain if parallelized.
- The many-to-many problem is again a network flow problem, and it can be solved by the network simplex algorithm (or other specialized minimum cost network flow problems).


**Strengths And Weaknesses:**

**Strengths**:
* The idea of using Wasserstein distances for computing independence test is interesting and relevant to the ML community (but it is not novel)
* The proposed algorithm is better than a naïve application of the Hungarian algorithm
* The computational results look reproducible by a knowledgeable and coding-skilled reader

**Weakness**:
* The paper has flaws in the presentation and writing. They never cite the semi-assignment problem, which was studied in several papers from the ’90, see for example:

1. Jonker, R. and Volgenant, A., 1987. A shortest augmenting path algorithm for dense and sparse linear assignment problems. Computing, 38(4), pp.325-340.

2. Kennington, J. and Wang, Z., 1992. A shortest augmenting path algorithm for the semi-assignment problem. Operations Research, 40(1), pp.178-187.

3. Balas, E., Miller, D., Pekny, J. and Toth, P., 1991. A parallel shortest augmenting path algorithm for the assignment problem. Journal of the ACM (JACM), 38(4), pp.985-1004.

* The presentation at page 2 in the introduction is imprecise. The problem the authors are presenting is a semi-assignment problem, which is a special case of minimum cost flow problem, and which can be solved by linear programming using simplex-based algorithms (worst case exponential) and/or interior point algorithms (polynomially solvable). In particular, the minimum cost flow problem is strongly polynomially solvable by primal Network Simplex algorithms, see for example:
4. Orlin, J.B., 1997. A polynomial time primal network simplex algorithm for minimum cost flows. Mathematical Programming, 78(2), pp.109-129.

* In addition, interior point algorithms were customized for bipartite min cost flow problems:

5. Resende, M.G. and Veiga, G., 1993. An implementation of the dual affine scaling algorithm for minimum-cost flow on bipartite uncapacitated networks. SIAM Journal on Optimization, 3(3), pp.516-537.

* About the references on Optimal Transport algorithms (i.e. “Approximation algorithms” in related work section), it would be better to distinguish among algorithms that compute the optimal solution value and the optimal decision variables values (i.e., the optimal transportation plan), and the algorithms that compute or approximate the optimal solution value without actually computing a feasible transportation plan. In the two cases, the (pseudo)polynomial worst-case complexity is different, with the second type of algorithm having a better worst-case complexity.

* The list of definitions in section 2 can be improved for clarity. For notations and definitions of these types of networks flow problems, we suggest referring to:
6. Ahuja, R.K., Magnanti, T.L. and Orlin, J.B., 1988. Network flows.

* The instances used in the benchmark are not very large, and we would suggest also using the standard DOT-benchmark to different image resolutions to study the scalability of the proposed approach:
7. Schrieber, J., Schuhmacher, D. and Gottschlich, C., 2016. Dotmark–a benchmark for discrete optimal transport. IEEE Access, 5, pp.271-282.

---

> ### Author Response · Authors · 2023-01-17
> **Response to Reviewer ayht**
>
> $\bullet$ **``You should clearly explain how your algorithm differs from the Kennington, J. and Wang, Z., 1992 algorithm because it does not look too different.''**
>
> The proposed modified Hungarian algorithm is different from the algorithm proposed in [3], because they are based on different algorithms.
> Our algorithm modifies the Hungarian algorithm [4], while the algorithm in [3] adjusts the shortest path augmenting algorithm [1, 2].
> As illustrated in [2, 3], the Hungarian algorithm is a primal-dual method based on maximum flow, while the shortest path augmenting algorithm considers the assignment problem as a minimum cost flow problem and is a dual method based on the shortest path.
> More specifically, the modified Hungarian algorithm is based on a modified Kuhn-Munkres theorem and iterates to identify a perfect  pseudo-matching in a bipartite graph with some feasible dual variables, while
> the algorithm in [3] involves constructing the shortest augmenting path in the auxiliary graph, and the flow is pushed along the path [3].
> In terms of computational complexity, both algorithms have the order of $\mathcal{O}(m^2n)$.
> However, our algorithm is easier to understand and implement because there are four phases (including column reduction, reduction transfer, row reduction augmentation, and shortest path augmentation) in the algorithm in [3], while our algorithm only involves two phases (updating the feasible labeling and improve the pseudo-matching).
>
> We have added this comparison in Section 1.1.1 in the revised paper.
>
> $\bullet$ **``...In particular, the minimum cost flow problem is strongly polynomially solvable by primal Network Simplex algorithms''**
>
> We make a theoretical comparison with the network simplex algorithm.
> Note that [5] proposes the first polynomial-time network simplex algorithm, and [6] further improves the result. The associated computational complexity of applying Tarjan's algorithm to the OT problem is $\mathcal{O}(m^2n\log(m)\min\{ \log(m C_{max}),mn\log (m)\})$, where $C_{max}$ denotes the maximum absolute value of the costs if all costs are integers and $\infty$ otherwise [5,6].  More specifically, if the costs are integral, the resulting computational complexity is $\mathcal{O}(m^2n\log(m)\log(m C_{max}))$, which is comparable to the proposed algorithm; if the costs are not integral, the resulting computational complexity is $\mathcal{O}(m^3n^2\log^2(m))$, which is much worse than our algorithm.
> In conclusion, fast computation requires integer-valued costs, while the proposed modified Hungarian algorithm could handle real-valued costs.
> Although most of the problems in practice have rational costs, which can be scaled to integer-valued costs, it is more convenient to use the proposed algorithm since it allows direct input of any real-valued costs. Also, looking into OT problems with real-valued costs itself is of much theoretical interest.
>
> We have added this comparison in Section 1.1.3 in the revised paper.
>
> $\bullet$ **``..., at least they should compare with the solution of the semi-assignment problem via the Network Simplex as implemented in the COIN-OR Lemon Graph library....''**
>
> We make a numerical comparison with the network simplex algorithm in Section 6 of the revised paper.
> Because Lemon Graph library is mainly based on C++ while our modified Hungarian algorithm is implemented in Python, we run the network simplex method from the networkX library in Python instead of Lemon Graph library. The experiment results can be found in Figures 15-18 in the revised paper. The figures show that the modified Hungarian algorithm compares favorably to the network simplex method.
>
>
> [1] E. Balas, D. Miller, J. Pekny, and P. Toth. A parallel shortest augmenting path algorithm for the assignment problem. Journal of the ACM (JACM), 38(4):985–1004, 1991.
>
> [2] R. Jonker and A. Volgenant. A shortest augmenting path algorithm for dense and sparse linear assignment problems. Computing, 38(4):325–340, 1987.
>
> [3] J. Kennington and Z. Wang. A shortest augmenting path algorithm for the semi-assignment problem. Operations Research, 40(1):178–187, 1992.
>
> [4] H. W. Kuhn. The hungarian method for the assignment problem. Naval research logistics quarterly, 2(1-2):83–97, 1955.
>
> [5] J. B. Orlin. A polynomial time primal network simplex algorithm for minimum cost flows. Mathematical Programming, 78(2):109–129, 1997.
>
> [6] R. E. Tarjan. Dynamic trees as search trees via euler tours, applied to the network simplex algorithm. Mathematical Programming, 78(2):169–177, 1997.

---

> ### Author Response · Authors · 2023-01-17
> **Continued Response to Reviewer ayht**
>
> $\bullet$  **``In addition, interior point algorithms were customized for bipartite min cost flow problems''**
>
> As mentioned by the reviewer, the interior point algorithms can be customized to solve the minimum-cost flow
> problems [7].
> We make a comparison with the interior point method.
> As discussed in [8], from the perspective of computational complexity,
> [10] proposes a strongly polynomial-time interior point algorithm for solving the minimum-cost flow problem.
> The adaption of this method to the OT problem has the order of $\mathcal{O}(m^{6.5}n^{6.5}\log (mn))$, which is much worse than the proposed algorithm.  As demonstrated in [8], prior to  [10], to solve the minimum-cost flow problem, the fastest interior point method comes from [9]. The
>  computational complexity of the adoption of Vaidya's algorithm to the OT problem is $\mathcal{O}(m^{2.5}n^{0.5}\log(mC_{\max}))$. The complexity is worse than our algorithm, and the algorithm requires that all costs are integers.
>
>  We have added this comparison in Section 1.1.3 in the revised paper.
>
> $\bullet$ **``About the references on Optimal Transport algorithms, it would be better to distinguish among algorithms that compute the optimal solution value and the optimal decision variables values, and the algorithms that compute or approximate the optimal solution value without actually computing a feasible transportation plan...."**
>
> The reviewer states that there may exist two categories of approximate algorithms to solve OT problems: (I) approximate both the optimal value and the optimal decision; (II) approximate the optimal value without computing a feasible plan.
> However, all approximation algorithms mentioned in our paper belong to category (I), i.e., approximating both the optimal value and the optimal decision.
> More specifically,
> recall the optimal transport (OT) problem:
> $\min_{X\in \mathcal{U}(\alpha,\beta)}\langle C,X\rangle,  \mathcal{U}(\alpha,\beta):=\\{ X\in \mathbb {R}^{n\times n}_{+}| X \textbf{1}_n=\alpha, X^{T}\textbf{1}_n=\beta \\}.$ All  approximation algorithms mentioned in our paper are to obtain  an $\epsilon$-approximation solution $\widehat{X}\in\mathcal{U}(\alpha,\beta)$ to the OT problem, such that  $\langle \widehat{X},C\rangle \leq \langle X^{\ast},C\rangle +\epsilon, $
> where $X^{\ast}$ is the solution to the OT problem.
>
>
> $\bullet$ **``The instances used in the benchmark are not very large, and we would suggest also using the standard DOT-benchmark to different image resolutions to study the scalability of the proposed approach''**
>
> We have added numerical experiments on the standard DOT-benchmark in the revised paper. The results illustrate that our modified Hungarian algorithm performs well in the large-scale dataset.
>
>
> $\bullet$ **Section 4 does not add too much value to the paper, and it could be merged with the introduction.**
>
> We have merged Section 4 with the Introduction and the associated complexity analysis in Section 3 in the revised paper.
>
> $\bullet$ **``We suggest including the plots of the computational results with the running time in the paper, and postponing in the appendix the results with the number of operations...''**
>
> We have included the numerical results with running time in the main body of our paper. But we keep results of numerical operation when we compare with the  Hungarian algorithm for the validation of the theoretical analysis of the computational complexity.
>
>
> $\bullet$ **``The list of definitions in section 2 can be improved for clarity.''**
>
> We have improved this part as suggested in the revised paper.
>
>
> $\bullet$ **``The many-to-many problem is again a network flow problem, and it can be solved by the network simplex algorithm...''**
>
> We agree with the reviewer's  comments that the many-to-many problem (4) is again a minimum-cost flow problem and can be solved by the exact solvers, including the network simplex, the interior point method, etc. Among the algorithms, the computational complexity of the network simplex algorithm can be comparable to the proposed modified Hungarian algorithm---$\mathcal{O}(M^2n)$---under the assumption that all costs are integral.
>
> We have added this discussion in Section 1.1.3 in the revised paper.
>
> $\bullet$ **``...in the definition of $\Pi^\circ$ is unclear the quantification over the constraints (for all $i,j,k$?)
> In problem (2), are the $m_j$ variable or parameters? ...''**
>
> We have modified these parts in the revised paper.
>
> [7] M. G. Resende and G. Veiga. An implementation of the dual affine scaling algorithm for minimum-
> cost flow on bipartite uncapacitated networks. 1993.
>
> [8] M. G. Resende and P. M. Pardalos. Interior point algorithms for network flow problems. 1996
>
> [9] P. M. Vaidya. Speeding-up linear programming using fast matrix multiplication.  1989.
>
> [10] S. A. Vavasis and Y. Ye. An accelerated interior point method whose running time depends only
> on A. 1994.

---

### Review · Reviewer_zbUw · 2023-01-04

**Summary Of Contributions:**

This paper introduces a modified Hungarian algorithm to solve exactly Integer Linear Programs of the form described in Equation (3). These optimization problems are equivalent to find a pseudo-matching (Definition 8) on some equality graph (Theorem 1). The algorithm proceeds by iteratively improving the labeling and the pseudo-matching (Algorithm 1). The authors then detail 3 applications: an independence test using the Wasserstein distance, the one-to-many assignment problem, and the many-to-many assignment problem, for which the proposed algorithm improves over existing methods in terms of computational time. Some numerical experiments are presented, that confirm the computational superiority of the proposed algorithm (Section 7).

**Audience:**

Yes

**Broader Impact Concerns:**

No concerns

**Claims And Evidence:**

Yes

**Requested Changes:**

Address the weaknesses detailed above. In particular:
- clarify the exposition and detail the broadest setting in which the proposed algorithm can be applied
- provide reminders about the Hungarian algorithm to make the modification explicit
- clarify the difference between one-to-many and many-to-many assignment problems
- clarify the computational complexities
- clarify the experiments
- clarify the related work section

Other points:
- in the introduction $\gamma(\nu_1, \nu_2)$ is never defined, nor $\hat{\nu}_1$ and $\nu_2$. This should be clarified
- Proposition 2 should appear explicitly in the text
- the assignment problem should be defined when discussed right after eq.(3), or at least a pointer given

**Strengths And Weaknesses:**

**Strengths**
- despite the exposition (that needs to be re-organized in my opinion), the paper is globally clear and well written
- the computational improvement is significant
- the topic is of interest to the TMLR community

**Weaknesses**
- as mentioned above, I feel that the organization of this paper should be revised. If I can understand that the Wassertstein independence test is one of the main motivations to the present contribution, I would not center the exposition around it. In particular, by Section 4 it seems that the proposed algorithm can be applied to a class of OT problems that is way more general than independence tests. For instance, if we have two empirical distributions composed respectively of $m$ and $n$ atoms, cannot we choose $M = mn$, $n_i=m$ and $m_j=n$, such that computing the Wasserstein distance between the two distributions is actually a special case of Problem (4)?
- I think it would benefit the paper to provide some reminders on the non-modified Hungarian algorithm, for the reader to be able to appreciate the modification and see where it plays a role in reducing computations. This point is far from being clear at the moment.
- the one-to-many and many-to-many assignment problems are presented as two different applications although the latter seems to be a strict generalization of the first one, or am I missing something here?
- in Example 1, the computation complexities $\mathcal{O}(4m_1^2)$ and $\mathcal{O}(5m_1^2)$ do not make sense, as 4 and 5 are constants already taken into account by the $\mathcal{O}$ notation. Instead, it would make more sense to present a generalized setting with an arbitrary number $r$ of roles and present computational complexities that depend on $r$.
- regarding the experiments, why do the authors present experiments with and without dependence? Is it supposed to influence the algorithm's speed? Same question for $p$. It seems to me that once the cost matrix $C$ is computed, the algorithm is oblivious to anything else. Could the authors explain the big variations, when any, between the best and worst run? When benchmarking Sinkhorn, I would like to see the precision achieved as a function of the number of operations. Indeed, the algorithm might attain a reasonable precision very quickly, which makes it an interesting tool if one is not interested in solving the problem exactly.
- the related work section is very brief and do not provide any details about what is done in the cited articles

---

> ### Author Response · Authors · 2023-01-17
> **Response to Reviewer zbUw**
>
> We thank the reviewer for the careful comments, and we reply to the comments as follows:
>
>
> $\bullet$ **``clarify the exposition and detail the broadest setting in which the proposed algorithm can be applied''**
>
> As the reviewer discussed, problem (4) is the broadest setting to which our proposed algorithm can be applied. Problem (2) is a special case of problem (4), where $n_i=1, M=m,\forall i=1,\cdots,m$, and then the Wasserstein distance independence test is also a special case of problem (4), where
> $n_i=1, m_j=n, M=n^2, \forall i=1,\cdots m;j=1\cdots n.$
>
> To make the exposition clear,
> we have merged Section 4 to the Introduction and analyzed the computational complexity in Section 3.5 in the revised paper.
>
>
> The reasons why we center the exposition around the Wasserstein independence and focus on problem (2) are: firstly, the proposal of pseudo-matching and the associated modified Hungarian algorithm is originally inspired by the solution structure of problem (2); secondly, we could solve the general problem (4) by first converting it to problem (2) and then applying the proposed algorithm.
>
> $\bullet$ **``provide reminders about the Hungarian algorithm to make the modification explicit.''**
>
> We have added more reminders of the original Hungarian algorithm in Section 3.1 and some discussions of the modification in Section 3.6 in the revised paper.
>
>
>
> $\bullet$ **``clarify the difference between one-to-many and many-to-many assignment problems''**
>
> Yes, the one-to-many assignment problem could be considered as a generalization of the many-to-many assignment problem. More specifically, one-to-many assignment problem corresponds to problem (2) while many-to-many assignment problem corresponds to problem (4).
>
> We have added this clarification in Section 5 of the revised paper.
>
> $\bullet$ **``clarify the computational complexities''**
>
> If there are $r$ roles, the associated computational complexities are $\mathcal{O}(rm_1^2)$ and $\mathcal{O}((r+1)m_1^2)$, respectively.
>
> We have modified the associated part in the revised paper.
>
> $\bullet$ **``clarify the experiments.''**
>
> We agree with the reviewer's comments that once the cost matrix is computed, the algorithm is oblivious to anything else.
> Notably, the cost matrix depends on the samples we select and the value of $p$. The purpose of presenting the experiments with and without dependence is to create different kinds of sample sets.
> In each experiment, we randomly choose $n$ samples from the sample set and compute the associated cost matrix based on the $l_p$ norm.
> In this way, we can test the efficiency of the proposed algorithm for different instances, i.e., different cost matrices.
>
> The variations of the numerical operations/time come from the variations of the cost matrices. Because the cost matrices are different from each other in our experiments, the algorithms run in different ways and then have different computational complexities.
> More specifically, the modified Hungarian algorithm initiates a pseudo-matching in the first step and then iterates to augment the pseudo-matching to be a perfect one. In some instances, the structure of the cost matrix enables the initialization step to give an almost perfect pseudo-matching, and it will take less effort to augment it. In this case, the associated computational complexity will be smaller. In other instances, however, the initialization may give a pseudo-matching that is far from perfect. In this case, the associated computational complexity will be larger.
>
> We admit that Sinkhorn can attain reasonable precision very quickly and is an interesting tool if one is not interested in solving the problem exactly.
> In our experiments, we set the accuracy for Sinkhorn as 0.0001 and want to demonstrate that our algorithm will outperform if one is interested in exact solutions or solutions with high accuracy. We have added this claim accordingly in Section 6.3 in the revised paper.
>
> Further, from our perspective, studying the precision achieved as a function of the number of operations for the Sinkhorn algorithm is out of the scope of this paper. In addition, considering our algorithm is an exact algorithm, studying the precision is also not meaningful to our proposed algorithm.
>
> $\bullet$ **``clarify the related work section''**
>
> More details have been added in the Related work in the revised paper.
>
> $\bullet$ **Other points concerning the definitions and proposition 2**
>
> We have modified these parts accordingly in the revised paper.

---

### Review · Reviewer_F8ov · 2023-01-16

**Summary Of Contributions:**

The paper has a simple and clear contribution: a modification of Hungarian algorithm to solve a special class of optimal transport  (OT) problems. This is useful, for example, for the application of Wasserstein distance for independence test, reducing the computational complexity from $\mathcal O(n^6)$ to $\mathcal O(n^5)$. The special class of OT problems include the ones defined for empirical measures which can be equivalently expressed as a (pseudo) matching problem. The computational complexity is theoretically proven and illustrated on several numerical examples.

**Audience:**

Yes

**Claims And Evidence:**

Yes

**Requested Changes:**

Overall, I like the paper and I think it is making a solid contribution. My only minor request for change is about the generalization section. The proposed idea is to increase the number of atoms for one marginal from $n$ to $M$ by creating $n_i$ atoms for each node. This seems in contrast to the original motivation of the paper, which was about exactly not performing this action on the other marginal. There should be a better way to do this. So I am not a fan of calling this a "generalization" of your proposed approach.

**Strengths And Weaknesses:**

Strength:
- clear presentation of a simple but effective idea
- theoretical results along with numerical experiments
- comparison with related work

Weakness:
- the generalization section might be a bit confusing

---

> ### Author Response · Authors · 2023-01-17
> **Response to Reviewer F8ov**
>
> Great thanks for the reviewer's appreciation and careful comments for our paper!
>
> We agree with the reviewer's comment that the content in Section 5 may not be considered as a 'generalization'.
> To avoid the confusion, we have deleted Section 5 and merged the content of Section 5 to the Introduction and Section 3.

---

### Author Response · Authors · 2023-01-17
**Revised paper**

Dear Action Editor and Reviewers,

Thank you for your constructive comments on our work! We have uploaded the revised paper according to your comments. The modifications are highlighted in red.

There are two main aspects of revision:
(1) We make a more comprehensive literature review and make a more careful comparison with the the existing work.
(2) To avoid the confusion, we delete the 'generalization' Section and merge the content to the Introduction and Section 3.

Because the length of the revised paper is more than 12 pages, we choose 'Long submission'.

If there are other comments and concerns, please let us know!


Best Regards,
Authors

---

### Decision · Action_Editors · 2023-02-15

**Recommendation:** Accept as is

**Comment:**

The paper makes simple yet clear and concrete algorithmic contributions to a broad class of problems. There were some concerns initially regarding the presentation and organization of the paper and comparison with the related work. The authors have addressed them adequately in the revision they posted.

**Audience:**

Theoreticians and practitioners interested in algorithmic approaches to solving optimal transport problems.

**Claims And Evidence:**

The paper contributes a modification of the Hungarian algorithm to solve a class of optimal transport problems. Among other applications, this is useful in computing empirical Wasserstein distance exactly, reducing the computational complexity from O(n^6) for the classic Hungarian algorithm to O(n^5) with the proposed modification, where n is the number of atoms. In an empirical study supporting the theoretical results, the proposed algorithm was found to fare favorably against the classic Hungarian algorithm, the popular Sinkhorn algorithm, and the network simplex algorithm. Overall, a good paper.

---

> ### Author Response · Authors · 2023-02-20
> **Camera ready version**
>
> Dear AE and all reviewers,
>
> Thank you for the constructive feedback that helps improve the presentation and quality of the paper.
>
> We have uploaded the camera-ready version and associated code of our paper.
>
> Sincerely,
>
> Authors